

# Snow-darkening versus direct radiative effects of mineral dust aerosol on the Indian summer monsoon: role of the Tibetan Plateau

Zhengguo Shi[1, 2, 3], Xiaoning Xie[1], Xinzhou Li[1, 2], Liu Yang[1], Xiaoxun Xie[1], Jing Lei[1], Yingying Sha[1], and Xiaodong Liu[1, 2]

[1]State Key Laboratory of Loess and Quaternary Geology, Institute of Earth Environment, Chinese Academy of Sciences, Xi'an 710061, China
[2]CAS Center for Excellence in Tibetan Plateau Earth Sciences, Beijing 100101, China
[3]Open Studio for Oceanic-Continental Climate and Environment Changes, Qingdao National Laboratory for Marine Science and Technology, Qingdao, China

*Correspondence to:* Dr. Zhengguo Shi (shizg@ieecas.cn)

**Abstract.** Mineral dust aerosol exerts complicated effects on the climate system and two of which are through their direct radiative forcing and snow-darkening forcing. Especially, the snow-darkening effect of dust on climate have been scarcely explored till now. When depositing in snow, dust can reduce the albedo of snow by darkening it and increase the snow melt. In this study, the snow-darkening effect of dust, as well as the direct radiative effect, on the Indian summer monsoon are

5    evaluated by atmospheric general circulation model experiments, with a special focus on the role of Tibetan Plateau. The results show that, the snow-darkening and direct radiative forcing of dust have both significant impacts on the onset of Indian monsoon but they are distinctly opposite. The snow-darkening effect weakens the Indian monsoon precipitation during May and June while the direct radiative forcing intensifies it. The surface temperature over western Tibetan Plateau and Central Asia becomes warmer due to the dust-induced decrease in snow cover, which leads to a local low-level cyclonic anomaly as well as

10    an anticyclonic anomaly over Indian subcontinent and Arabian Sea. This circulation pattern allows air current penetrating into Indian subcontinent more from Central Asia but less from Indian Ocean. In contrast, the direct radiative forcing of dust cools Tibetan Plateau and adjacent areas but warms Arabian Peninsular, which intensifies the moisture convergence and upward motion over Indian monsoon region. The upper tropospheric atmospheric circulation over Asia is also sensitive to both effects. Our results highlight a potential role of snow-albedo feedback in the effects of dust, which significantly amplifies the response

15    of temperature over Tibetan Plateau. Thus, links between the climatic impact of dust, Tibetan Plateau thermal condition and surface snow cover are of importance and require to be clarified accurately.





## 1 Introduction

Mineral dust, a kind of natural aerosols in the atmosphere, mainly originates from the global deserts including Sahara, Arabian peninsula, Central Asia and East Asia. Dust emitting into the atmosphere is carried by atmospheric circulation and can be transported downwind for a long distance (Zhang et al., 1997; Zhao et al., 2006; Kallos et al., 2006; Schepanski et al., 2009;

Shi and Liu, 2011). Mineral dust aerosol affects global and regional energy budget, formation of clouds and precipitation as well as various climate systems through their direct, semi-direct and indirect effects (e.g., Tegen and Lacis, 1996; Ramanathan et al., 2001; Miller et al., 2004; Shao et al., 2011; Huang et al., 2014; Mahowald et al., 2014). Among the climatic effects of dust, the direct radiative effect (DRE) and snow-darkening effect (SDE) are two important components, which exert great impacts on the radiative balance (Haywood et al., 2001; Flanner et al., 2009; Huang et al., 2014; Qian et al., 2015).

The DRE of dust is that the particles can directly reflect, scatter and absorb the solar shortwave and black-body longwave radiation. In the fifth Assessment Report (IPCC, 2013), the annual mean DRE of dust is approximately $-0.10$ W m$^{-2}$ on the global scale, which varies from $-0.30$ to $+0.10$ W m$^{-2}$ among different global climate models. However, it is still unclear whether dust aerosol has a net warming or cooling effect on global climate (e.g., Tegen and Lacis, 1996; Miller and Tegen, 1998; Mahowald et al., 2014; Kok et al., 2017; Xie et al., 2018a). Due to the underestimation of coarser dust in climate models

than in the atmosphere, the considered DRE may be more cooling in current model ensemble and the possibility that dust causes a net warming is highlighted (Kok et al., 2017).

Following the changes in radiative balance, specific climate systems or atmospheric circulations also respond significantly to the DRE of aerosols. During the emission seasons, dust from inland Asian and Arabian deserts is delivered downwind by the westerlies and Asian monsoon (Uno et al., 2009; Shi and Liu, 2011; Vinoj et al., 2014) to eastern China, India and even

deposits in the Tibetan Plateau (Huang et al., 2007; Xu et al., 2009; Zhang et al., 2018). Such distributions of atmospheric dust largely affect the Asian climate, including both Indian and East Asian monsoon (Lau et al., 2006; Zhang et al., 2009; Sun et al., 2012; Vinoj et al., 2014; Jin et al., 2014; Gu et al., 2016; Lau et al., 2017; Lou et al., 2017). Via a strong effect of elevated heat pump, the DRE of absorbing aerosols including dust enhances the heat source over the TP and results in a northward shift of Indian summer monsoon during the late spring and early summer (Lau et al., 2006; Lau et al., 2017).

The aerosol-induced upper tropospheric warming intensifies the updraft air motion, which pumps more moist air from south oceans to north India. However, this hypothesis is still in debate that lacks of observational support (Nigam and Bollasina, 2010). Beside the TP warming, the tropospheric warming over Arabian Sea and surrounding regions due to mineral dust from Middle East can intensify the Indian summer monsoon and precipitation (Vinoj et al., 2014; Jin et al., 2014). In addition, the East Asian monsoon and the eastern precipitation are also significantly affected by dust that northeasterly wind anomaly over

eastern China seems to weaken the monsoon circulation (Sun et al., 2012; Tang et al., 2018).

SDE is another important effect of dust on climate, which is not mentioned as intensively as the DRE. Light absorbing aerosols can darken the snow and reduce the surface albedo when deposited in snow, and it can also absorb the radiation and warm the snow surface, which both accelerates the melt process of snowpack (Hansen and Nazarenko, 2004; Xu et al., 2009; Lee et al., 2013; Qian et al., 2015). Due to the reduction of snow, the SDE of absorbing aerosols generally induces a net regional




warming over the snow cover areas. Black carbon, as the most important anthropogenic absorbing aerosols, has a global-mean radiative forcing of +0.04 (+0.02 to +0.09) W m$^{-2}$ for SDE (Bond et al., 2013). Over the Tibetan Plateau (TP) where most areas are covered by snow, in particular, the absorbing aerosols in snow remarkably influence the snow albedo and promote the snowmelt (Lau et al., 2010; Yasunari et al., 2011). The SDE of black carbon generates positive changes in surface radiative

flux of about $5-25$ W m$^{-2}$ over the TP during springtime, warms the surface TP obviously and intensifies both the Indian and East Asian summer monsoon by enhancing the elevated heat source (Qian et al., 2011; Qian et al., 2015).

Compared to that of black carbon, the SDE of mineral dust over TP and Asia, especially its influence on the Asian monsoon, is still not clear. Theoretically, the SDE of dust is considered to be larger than that of black carbon over the TP (Flanner et al., 2009; Ming et al. 2013) primarily because the concentration of dust is much higher. The spatial distribution and deposition of

dust is also different from black carbon that the dust can be deposited over both central Asia and TP where exists a fraction of snow cover while black carbon is primarily restricted to South and East Asia and downwind areas. In actual, the dust is a kind of natural aerosols, differing from black carbon which is mainly anthropogenic produced. Beside the modern period, the climatic effect of Asian dust are also of great importance in the geological stages, such as the last glacial maximum (Harrison et al., 2001; Claquin et al., 2003; Takemura et al., 2009). During the late Cenozoic, the dust effect ought to become gradually

larger as deserts expand and atmospheric dust increases with plateau uplift and climatic cooling (Shi et al., 2011). Thus, it is necessary to explore in detail the effect of dust during present day and geological periods.

In this paper, as a first step, we employed a set of numerical experiments by a general circulation model to evaluate the SDE and DRE of dust on Indian summer monsoon during the onset under present-day conditions. Potential role of TP thermal changes is specifically focused because there is a large fraction of snow cover over the plateau. In Section 2, the model and

experiments are described. The model performance, response of Indian monsoon and role of thermal Tibetan Plateau are presented in Section 3. The discussion and conclusions are summarized in Section 4 and 5, respectively.

## 2 Model and Experiments

An atmospheric general circulation model namely Community Atmosphere Model 4 (CAM4), which is improved with a new bulk aerosol model (BAM) parameterization, is employed to evaluate the response of Indian summer monsoon to the forcing

of mineral dust. CAM4 is the atmospheric component of the Community Climate System Model 4 (CCSM4), which is coupled with the Community Land Model 4 (CLM4) for land surface processes. The vertically Lagrangian and horizontally Eulerian coordinates are used in the finite-volume discretization of this model. The dust cycle including the emission, transport and deposition, is parameterized in CAM4 and its radiative feedbacks are also calculated on line. The original dust sizes in CAM4 contain four bins of $0.1-1.0$ $\mu$m, $1.0-2.5$ $\mu$m, $2.5-5.0$ $\mu$m and $5.0-10.0$ $\mu$m in diameters, respectively (Mahowald et al.,

2006). The CAM4-BAM has been improved by an optimized soil erodibility map and a new size distribution for dust emission, as well as updated optical properties for radiation budget, to present a better performance on simulating the global dust cycle (Albani et al., 2014). In CAM4-BAM, the SDE of all aerosols are enabled but the indirect effect is not considered, which means that the aerosol changes in cloud process as condensation nuclei are prescribed. The snow darkening processes are



considered based on the Snow, Ice and Aerosol Radiative (SNICAR) module (Flanner et al., 2007; 2009) in which the dust and black carbon aerosols are included. The SNICAR applies Mie scattering to particle mixture and a multi-layer radiative transfer approximation (Toon et al., 1989) to represent vertical inhomogeneity in the snow. The radiative transfer in the snow is affected by the vertical particle profile controlling by fresh snow and flushing with melt water when dust deposits on the surface. Dust

optical properties in snow were ranging from 0.88 to 0.99 with decreasing particle size (Flanner et al., 2009).

Three sensitivity experiments are conducted in this study to evaluate the SDE and DRE of mineral dust. Both the snow-darkening and direct radiative feedbacks of dust are turned on in the experiment namely EXP1 while only the direct radiative feedback is enabled in the experiment of EXP2. Neither effects are taken into consideration in the third experiment (EXP3). Thus, the differences in climate responses between EXP1 and EXP2, and between EXP2 and EXP3, are denoted as the SDE

and DRE, respectively. Of note is that the dust column loading over Asia is slightly larger by the on-line feedbacks when both two effects are enabled, compared to that when DRE is only enabled. However, the bias does not affect our discussion, which will be mentioned later in this work. The reason for the intensified dust cycle over Asia by SDE is analyzed in detail in a parallel study (Xie et al., 2018b). Other species of aerosols except mineral dust are neglected in these experiments to avoid the biases induced by their different spatial distributions in different experiments. The boundary conditions, including the sea

surface temperature and greenhouse gas concentrations, are kept as their modern values (The year 2000 AD). The sea surface temperature and sea ice is given from HadOIBI data and the atmospheric $CO_2$ concentration is set to 367 ppmv.

In these experiments, the horizontal resolution of CAM4-BAM is set to approximately $0.9° \times 1.25°$ in latitude and longitude. All the experiments are integrated for a total period of 21 years and the results of the last 15 years are analyzed. Both monthly and daily mean values of variables are outputted to examine the sensitivity of monsoon. The response of Indian monsoon

circulation and precipitation during May and June (i.e., the onset) is focused in this study since the monsoon onset is sensitive to the external forcing, especially to thermal changes over the TP.

## 3   Results

### 3.1   Model validation

Before the examination of monsoon response, the model's ability on simulating the climatology of dust aerosol optical depth

(AOD), snow cover and Indian monsoon during May and June in the experiment EXP1 is first evaluated using modern observation and reanalysis data. The distributions of the AOD and deposition flux of mineral dust in the model over Asia are shown (Figure 1). The maximal values of May-June mean dust AOD are found over the arid and semi-arid regions including the Sahara, Arabian Peninsula, Central Asian and East Asian deserts (Figure 1a). The AOD reaches above 0.2 over major source areas. This simulated pattern is similar with the Multi-angle Imaging SpectroRadiometer (MISR) retrieved AOD over

the deserts (Figure 1b), which indicates that CAM4-BAM has a good performance on the dust cycle. The simulated absolute values of AOD over Arabian Peninsular and Taklimakan desert are biased low. Over India and eastern China where the industrial pollution is heavy, the observed AOD is larger because it is mainly composed of sulfate, organic and black carbon which is not considered in this study. The total deposition fluxes during March-April and May-June (Figures 1c, 1d) show that there are



remarkable dust depositions over Asia and adjacent oceans in both periods. In March and April, the dust deposition over East and Central Asian deserts and downwind regions is larger than that in May and June. In contrast, the deposition over Arabian Peninsula is more obvious in May and June, which is also detected over Arabian Sea and western Indian continent. Over the western and northeastern Tibetan Plateau (TP), the deposition flux is simulated with a range of about 0.02-0.16 $kg/m^2/yr$.

The simulated snow cover fractions over Asia during May and June show that surface snow exists over Central Asia, East Asia and the whole TP, with largest fractions over western TP (Figure 2a). In Moderate Resolution Imaging Spectroradiometer (MODIS) data, the observed snow cover is found over the same regions that maximal values are located around Caspian Sea, Mongolia, western and southeastern TP (Figure 2b), which is qualitatively consistent with that in the EXP1 simulation. Over the western TP, the MODIS observation presents a fraction larger than 80% but the simulated fraction is smaller. In particular,

the model underestimates the elevations of finer-scale mountains and corresponding snow cover fractions due to the coarser resolution, e.g., over the Tianshan mountains. The dust deposition in the surface snow over Asia and especially over western TP implies a potential influence on surface snow.

For the Indian monsoon climatology, a feature that the monsoon westerly winds are divided into two branches, with the northern one from Central Asian dry regions and southern one from moist Indian Ocean, is simulated in the 850hPa winds

during May and June (Figure 3a). During the monsoon onset, the southerly winds over this region gradually develop from the south to the north. During the same period, the Indian monsoon precipitation is mainly produced over the western sides of the Indian and Indo-China peninsulas as well as the southern slope of the TP (Figure 3c). These features of Indian monsoon circulation and precipitation are generally in agreement with the National Centers for Environmental Prediction/National Center for Atmospheric Research (NCEP/NCAR) reanalysis and Tropical Rainfall Measuring Mission (TRMM) satellite-retrieved data

(Figures 3b, 3d). Compared to the observations, the simulated precipitation is lighter over the western sides of two peninsulas but heavier over the southern slope of the TP. In brief, CAM4-BAM performs well in both the monsoon climatology and dust cycle over Asia, which builds confidence for assessing the climate sensitivity to dust forcing.

## 3.2   Response of Indian monsoon

The daily precipitation differences during May and June between EXP1 and EXP2, as well as between EXP2 and EXP3,

are calculated to examine the responses of monsoon onset to SDE and DRE (Figure 4). It is clearly seen that in all three experiments the precipitation rates over Indian monsoon area increase abruptly by an amount of approximately 10 mm $day^{-1}$ during several weeks in the onset (Figure 4a). In this 60-day period, the EXP1-EXP2 difference is mostly negative while the EXP2-EXP3 difference is positive (Figure 4b), which means that the SDE tends to weaken the Indian summer monsoon but the DRE likes to intensify it. This is also the reason why we choose May and June as the monsoon onset in the following

analysis. The SDE-induced precipitation decrease exceeds the DRE-induced increase in June, which results in a net reduction in precipitation; however, these two effects almost counteracts by each other and the total precipitation change in May is not significant.

The spatial distributions of May-June mean precipitation show that the precipitation rate is decreased by the SDE over most Indian monsoon regions and a remarkable difference by 1 mm $day^{-1}$ is detected over India (Figure 5a). Other regions with





statistically-significant precipitation changes are found over western and southeastern TP, parts of Central Asia and northeastern Africa. For DRE-induced response, the precipitation is promoted over Indian peninsula, Arabian Sea and Central Asia but suppressed over Bay of Bengal and southeastern TP (Figure 5b). Thus, the responses of Indian monsoon precipitation to the SDE and DRE are distinctly different during the onset, which highlights the complicated influence of mineral dust. The surface

temperature becomes warmer over most Asia, which responds to the SDE (Figure 5c). The most obvious warming, with an amplitude of larger than $1°C$ , is found over the whole western TP, parts of eastern TP and Mongolia where the surface snow cover is larger, which indicates that the SDE is significant at these regions. Another warming center is around Caspian Sea in Central Asia also with certain snow covers at this time. In contrast, the surface temperature difference induced by the DRE is significantly negative over the whole TP and northeastern India (Figure 5d). However, it is simulated to be warming over

Arabian Peninsula, which amplifies the zonal thermal gradient over Indian monsoon region.

     The responses of Indian monsoon circulation to the SDE and DRE are examined by the differences in 850 hPa wind vectors between experiments (Figures 6a, 6b). In the EXP1-EXP2 difference, a significant cyclonic anomaly is simulated over western TP and to its west there is also a cyclonic anomaly around the Caspian Sea although it is not clear (Figure 6a). The western TP cyclonic anomaly tends to intensify the northern branch of Indian monsoon westerly, allowing more dry air from Central

Asia penetrating into the monsoon region. However, the southern branch of the monsoon westerly is significantly decreased, which weakens the moisture transport from oceans in the south. This circulation anomaly over monsoon area agrees well with the simulated lighter precipitation, which supports that the Indian summer monsoon is weakened by the SDE during its onset. In addition, the westerly winds become stronger to the north of the TP, which might affect the dust emission further over that region. In the EXP2-EXP3 difference, the situation is quite different that the northern branch of monsoon westerly is

remarkably reduced in its intensity across the southern slope of the TP, the Persian Gulf and the northern Arabian Peninsula (Figure 6b). The westerly winds are also decreased over the Bay of Bengal and Indo-China Peninsula, however, it brings water vapor to the Indian Peninsula. The southern branch of Indian monsoon westerly over Arabian Sea is simulated to be stronger although in most regions it is not statistically-significant. The differences in the moisture convergence induced by the SDE and DRE show that the water vapors converge and diverge over most Indian monsoon region, respectively (Figures 6c, 6d),

consistent with the responses of precipitation (Figures 5a, 5b).

     The responses of Indian monsoon system in the mid- and high-troposphere are also examined (Figures 7, 8) because the anomalous heating center over the TP as well as the high pressure cell are both important for the monsoon development. On 500 hPa isobaric level, the TP warming anomaly is found strongest and the Central Asian warming is also significant (Figure 7a), which corresponds well to the surface warming at these regions due to the SDE (Figure 5c). Two cooling centers are

located over northern Arabian Peninsula and Bay of Bengal, respectively, in which the latter one might be associated with changes in the latent heat release. The strong signal of warming over the TP leads to a decrease in the geopotential height (Figure 7c) and a cyclonic anomaly locally (Figure 7e). The increase in temperature and geopotential height over Central Asia and the decrease over Mongolia (Figures 7a, 7c) indicate that the amplified zonal thermal gradient promotes the intensification of the westerly winds to the north of the TP (Figure 7e). An anomalous northerly is originated from the TP and blows to the

Bay of Bengal (Figure 7e), similar with that in the low-level atmosphere (Figure 6a).



For the DRE-induced response, the 500 hPa temperature presents an opposite pattern with that of SDE, in which a cooling maximum is located over the TP and a warming maximum is over Arabian Peninsula (Figure 7b). This Arabian warming significantly expands and shifts northwards compared to surface temperature. Following the temperature changes, those in geopotential height also show a dipole pattern (Figure 7d), which produces an anticyclonic anomaly over Arabian Peninsula

and Iran, and a cyclonic anomaly over northern India, respectively (Figure 7f). To the north of the TP, the westerly winds are weakened by the DRE, which might help the long-distance transport of mineral dust over Asia.

In the high-level atmosphere, the atmospheric responses to both forcing are almost reversed, as seen in the 200 hPa climatology (Figure 8). A SDE-induced dipole pattern of meridional temperature changes over Central Asia and TP (Figure 8a) results in a western weakening and a eastern strengthening of South Asian high pressure cell, i.e., a eastward shift of high

pressure cell (Figure 8c). In contrast, the opposite dipole temperature changes caused by DRE make the high pressure cell move westward (Figures 8b, 8d). Differences in 200 hPa wind vectors also show a couple of reversed circulation changes of cyclonic/anticyclonic cell in the west and anticyclonic/cyclonic cell in the east responding to the SDE and DRE, respectively (Figures 8e, 8f), which is consistent with the temperature and pressure changes. In addition, the SDE-associated increased southerly winds are simulated significant over Bay of Bengal, different from those in 500 hPa winds.

Changes in vertical motion show that low tropospheric subsidence occurs over most of the monsoon areas with the SDE but the DRE leads to ascending motion over Arabian Sea and western India (Figures 9a, 9b). In the middle troposphere, strong ascending motion is found over the TP (Figure 9c), which is closely linked with local surface warming by SDE (Figure 5c). In contrast, the subsidence dominates the adjacent areas outside the TP including the Indian and Indo-China peninsulas, as well as regions to the west and north of the TP (Figure 9c). For the DRE, the surface cooling produces local subsidence over the

TP while the ascending motion is presented to the north of the TP and northern India (Figure 9d). The spatial distributions of anomalous vertical motion over Indian monsoon region are in qualitatively coincidence with the simulated precipitation changes by SDE and DRE, respectively. Such circulation changes is also clearly seen in the cross sections for vertical versus meridional winds (Figure 10). Above the TP surface including the top and southern slope, the anomalous warming and cooling by SDE and DRE lead to local ascending and subsiding flows, respectively. These flows are not so strong and merely dominate

in the mid troposphere. However, compensatory subsidence for SDE and ascending motion for DRE, located to both the southern and northern slopes of the TP, are remarkable throughout the troposphere.

From the analysis above, in brief, the suppressed and increased monsoon precipitation during May and June are fundamentally resulted from the SDE and DRE induced changes in atmospheric thermal structure, respectively, especially over the low-level atmosphere where most mineral dust exists. The strongest signals for surface temperature changes in both scenarios,

as well as the sensible responses of circulation structure are found over the TP (Figures 5-10), highlighting the potential thermal role of TP in the climatic effects of mineral dust.

### 3.3   Role of Tibetan Plateau

In this section, we have analyzed the forcing mechanism of both SDE and DRE on the temperature changes, including the radiative budget and feedback of TP snow cover, to explore how the TP and other key places play roles in the mechanism.



The SDE on atmospheric radiation budget is mainly realized by the surface albedo change led by darkening the snow and accelerating the snowmelt as a kind of absorbing aerosols. A positive feedback loop of SDE is that the net incoming solar shortwave radiation flux increases at the surface when snow albedo is reduced, and the rising temperature further enhances the snowmelt and reduces the surface albedo. The SDE-induced differences in longwave and shortwave radiation fluxes for all-sky

conditions during May and June at the top of atmosphere (TOA), at the surface and in the column atmosphere is shown in Figure 11, respectively.

For both the TOA and the surface, the primary forcing of SDE is via shortwave radiation change since it is albedo-induced. Due to large snow cover, the strongest shortwave radiation change is found positive over western TP with a mean value of larger than 25 W/m$^2$ (Figures 11b, 11e), which indicates that both the TOA and the surface receive more shortwave radiation

while the scattering becomes less. The positive shortwave forcing near Indian Peninsula, not so strong as that over TP, is offset by the negative longwave one (Figures 11a, 11d), in which these changes should be associated with internal adjustment of climate, e.g., the water vapor change. As a result, the SDE totally means a positive net radiative forcing over western TP at the TOA and the surface (Figures 11c, 11f), which is the reason for local surface warming (Figure 5c). Additionally, the net surface radiative forcing is also positive and statistically significant to the south of Caspian Sea although its absolute value is not as

large as that over western TP (Figure 11f). For the column atmosphere, the shortwave radiation flux does not vary, supporting that the slight dust loading difference between EXP1 and EXP2 merely presents negligible radiation changes. The negative longwave radiation difference is merely found near Indian Peninsula (Figures 11g, 11h, 11i), indicating that the atmosphere loses energy over this region.

For the DRE, the radiative forcing is characterized by positive longwave and negative shortwave radiation differences at

both the TOA and the surface (Figures 12a, 12b, 12d, 12e) owing to the absorbing and scattering of radiation by dust aerosol. However, the TOA changes are less evident than the surface changes because the dust aerosol is primarily distributed in the low level. Notably, a significant difference of larger than 25W/m$^2$ in shortwave radiation is seen over western TP (Figures 12b, 12e), highlighting the potential feedback. This negative forcing of dust over western TP is also an important feature in net changes (Figures 12c, 12f). Further, the positive net TOA forcing is obvious over Arabian Peninsula (the net surface forcing is

also positive but not statistically significant), which indicates that the pattern of surface temperature change by the DRE (Figure 5d) is more likely controlled by the TOA radiation change. The net TOA and surface radiative forcing of DRE shares a distinct opposite pattern with that of SDE, which is responsible for different response of atmospheric temperature and Indian monsoon. As absorbing aerosol, the longwave and shortwave forcing for the column atmosphere is negative and positive, respectively (Figures 12g, 12h), with maximal values distributed over the large dust AOD region (Figure 1a). The positive net total forcing

of dust is found remarkable near Arabian Peninsula but not so large over East Asia (Figure 12i).

The strong responses of shortwave radiation over western TP in both SDE and DRE is directly linked to changes in the snow cover and surface albedo (Figure 13), which significantly affect the reflection of the incoming solar radiation. In the SDE-induced difference, the surface albedo is reduced by about 0.10-0.25 (Figure 13a), owing to the decreased snow cover fraction of nearly 20% (Figure 13c). The DRE is also amplified by the TP snow-albedo feedback, although the largest radiative

forcing is not over TP. The increased snow cover fraction by DRE leads to a larger albedo (Figures 13b, 13d), which makes



the surface cooling over the TP more obvious than those over source region (Figure 5d). Thus, the simulated TP temperature change, which remarkably influences the response of Indian monsoon, is actually the combined result of dust forcing and snow-albedo feedback.

## 4    Discussion

The physical mechanisms for SDE and DRE of mineral dust on the Indian summer monsoon during the onset are summarized by schematic diagrams, respectively (Figure 14). The initial forcing of SDE is occurs over western TP and Central Asia, which becomes warmer due to the darkened and decreased snow cover. Subsequently, two anomalous surface low pressure centers are produced and upward air flow dominates over these areas (Figure 14a). To their south, a forced high pressure and anticyclone anomaly is found over Bay of Bengal and India where the subsidence suppresses the formation of monsoon rainfall. For the circulation, the anticyclone strengthens the westerly air flow from the dry Central Asia but limits that from moist Indian Ocean. In contrast, the forcing of DRE, which ought to be obvious over deserts where the dust AOD is large, is actually largest over western TP under the impact of snow-albedo effect. The western TP cooling, as well as the DRE-induced warming over Arabian Peninsula, produce a couple of surface high and low pressure anomalies (Figure 14b). This production process is also amplified by the cooling-induced subsidence over TP and compensatory upward flow to the south. Such a pattern gives a SDE-opposite impact of circulation, which intensifies the cross-equatorial southerly and weakens the dry air flows from the north. As a result of compensatory effect, the downward and upward flows are simulated to the northwest of TP for SDE and DRE, respectively, which significantly affects the inland arid climate during that period.

The radiative forcing and remarkable TP warming at the surface and high troposphere, as a direct response to SDE of dust or other absorbing aerosols (e.g., black carbon), is also found in previous studies (Flanner et al., 2009; Lau et al., 2010; Qian et al., 2011). They proposed that the snowmelt process is rapid and efficient during the late spring and early summer (Lau et al., 2010; Qian et al., 2011; Qian et al., 2015) and this sensitive response of snow cover to SDE in melting season supports its significant role in Indian monsoon development simulated in this study. However, the response of Indian monsoon to SDE of black carbon during the onset (Qian et al., 2011, hereafter Qian2011) is different from what we found here for dust. Qian2011 emphasized that the polluted snowpack by black carbon over TP warms the local surface and enhances the sensible heat flux, which results in a earlier onset of Indian monsoon and heavier precipitation over northern India. The opposite responses of monsoon originate from different locations of surface warming that the warming is just over the entire TP in Qian2011 but our warming centers are located in quite west, even to Central Asia. The snowmelt confined to the western TP is similar with a previous study (Lau et al., 2010). The westward shift of warming likely forces the southerly winds over India (See Figure 15c in Qian2011) to Arabian Peninsula in this study. Although the forcing on radiative budget of these two absorbing aerosols are similar, their distributions obviously differ. The black carbon, mainly emitted from the Industrial countries, is generally transport eastwards and scarcely into upwind Central Asia. These differences in surface warming by dust and black carbon are also simulated in the experiments by NASA Goddard Earth Observing System Model (Yasunari et al., 2014). In addition, the fraction of snow cover over TP is overestimated in Qian2011, as they pointed out in their paper, which may provide some



upper limits for SDE of black carbon. But in this study, the simulated snow fraction over TP is smaller than observed, which may cause the underestimation on SDE of dust.

The DRE-strengthened Indian summer monsoon in this paper is in qualitatively agreement with previous studies (Lau et al., 2006, hereafter Lau2006; Gu et al., 2016; Lau et al., 2017), in which either dust or black carbon, or both of them, is

included. However, the similar results may share different mechanisms. For example, the DRE-strengthened Indian summer monsoon in Lau2006 by both dust and black carbon is ascribed to an elevated heat pump (EHP) mechanism that the aerosols heat the southern slope of TP by absorbing the radiation and the hot air rises, which draws in moisture convergence over India. However, the EHP mechanism fails to be obvious when only mineral dust is considered here, because the DRE of dust only induces a remarkable surface cooling over TP during May and June not a warming as shown in Lau2006. We do not

make sure whether this TP warming in Lau2006 can be produced by dust only. Interestingly, there are consistent intensified upward air motion over southern slope of TP in these studies (Lau et al., 2006; Gu et al, 2016) although simulated responses of surface temperature depends closely on models. In this study, the upward motion is actually compensatory as a result of strong downward motion right over TP led by surface cooling because there is also not significant temperature change for vertical atmospheric levels (Figures 7d, 8b). Thus, this dust-intensified local upward air motion is an important reason and

responsible for the heavier monsoon precipitation. Worthy of being pointed out is that the DRE of dust on surface temperature is largely uncertain and depends closely on the size distributions, optical properties and etc (Kok et al., 2017), which restricts our accurate understanding of dust effect. Another important factor in this study is Arabian warming, which acts like a feedback on dust-monsoon interaction and drives moisture from southern oceans to Indian monsoon areas. This mechanism gains support from synoptic-scale researches (Vinoj et al., 2014; Jin et al., 2014), which emphasized the modulation of western African dust

on Indian monsoon rainfall. Similar with these studies, we also simulated a significant warming of the surface and upper atmosphere (Figures 5d, 7b) by the absorbing property of dust over Arabian Peninsula. The different performance of dust-induced radiative forcing and temperature changes over East Asia and northern Africa can be explained by different surface albedo background and particle sizes (Liu et al., 2008; Takemura et al., 2009; Su and Toon, 2011; Xie et al., 2018a).

The role of TP surface temperature amplified by snow-albedo feedback is found significant in CAM4-BAM, which highlights

the importance of TP in both SDE and DRE of dust. Change in thermal condition over surface TP, which acts as a heat source and exerts great sensible heat flux to atmosphere, are proved to be essential in the establishment of Indian summer monsoon (e.g., Yanai et al., 1992; Wu and Zhang, 1998). Furthermore, change in snow cover over TP can also obviously affect the Indian monsoon by modifying the thermal TP forcing (e.g., Vernekar et al., 1994; Senan et al., 2016) and those over different parts of TP may play different roles (Wang et al., 2017). Our results indicate that the TP snow cover change is sensitive to both

dust effects, even for the comparatively small radiative forcing of DRE over TP. Distinct responses of monsoon to snow cover change over different areas of TP in Qian2011 and our study support the spatially-complicated effect of thermal TP. We cannot deny the possibility that the simulated TP snow cover change might be model dependent, thus, this dust-snow-temperature-monsoon link over TP needs to be examined in future, especially by other parameterizations and observation data. Anyway, it is reasonable to expect that the response of Indian monsoon would be amplified if snow cover fraction over TP responds

obviously to dust effects.



## 5 Conclusions

In this study, significant responses of Indian summer monsoon, including both circulation and precipitation during the onset, are proposed to the SDE and DRE of mineral dust, which is closely associated with the snow-albedo feedback over the TP. The SDE and DRE of dust are found to exert different impacts on monsoon system due to opposite temperature changes over the TP, highlighting the complexity of climate effect of dust. The SDE/DRE reduces/increases snow cover and warms/cools the surface over western TP, which weakens/intensifies the monsoon development and precipitation during May and June. As the net result of SDE and DRE, the precipitation in June is reduced although dust effects are complicated. Beside the Indian monsoon, East Asian monsoon should be also affected by the dust-induced TP thermal change, which will be examined in future. Compared to other absorbing aerosols (e.g., black carbon) presenting positive TOA forcing, the DRE of dust on atmospheric radiation budget and thermal structure are still uncertain, which adds difficulty to evaluate the sensitivity of specific climate system to dust effects. Nevertheless, the role of dust still requires to be deeply explored due that it is natural and ought to be important during past climate change. In particular, several times larger dust burden and deposition during the Last Glacial Maximum (Mahowald et al., 2006; Maher et al., 2010), as well as higher snow cover fraction due to cold climate, are likely to induce stronger DRE and SDE than present day.

*Acknowledgements.* This work was jointly supported by National Key Research and Development Program of China (2016YFA0601904), the Strategic Priority Research Program of Chinese Academy of Sciences (XDA20070103) and the National Natural Science Foundation of China (41572160). Shi Z. also acknowledged the support of Youth Innovation Promotion Association CAS and "Light of West China" Program.




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




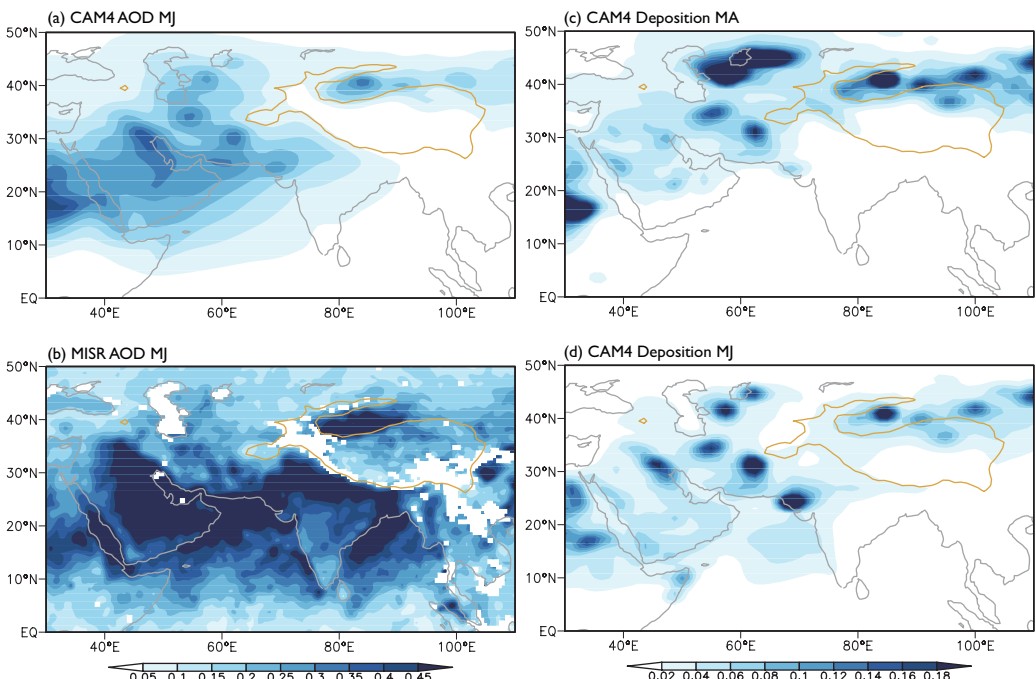

**Figure 1.** Averaged dust aerosol optical depth over Asia for May and June in CAM4 (a) and in Multi-angle Imaging SpectroRadiometer (MISR)-retrieved data (b); and mean dust deposition flux including both dry and wet deposition for March and April (c) (kg/m$^2$/yr) and for May and June (d). Yellow line shows the profile of Tibetan Plateau above 2500 m.



**Figure 2.** Snow cover fraction (%) over Asia and the Tibetan Plateau for May and June in CAM4 (a) and in Moderate Resolution Imaging Spectroradiometer (MODIS)-retrieved observation. Yellow line shows the profile of Tibetan Plateau above 2500 m.





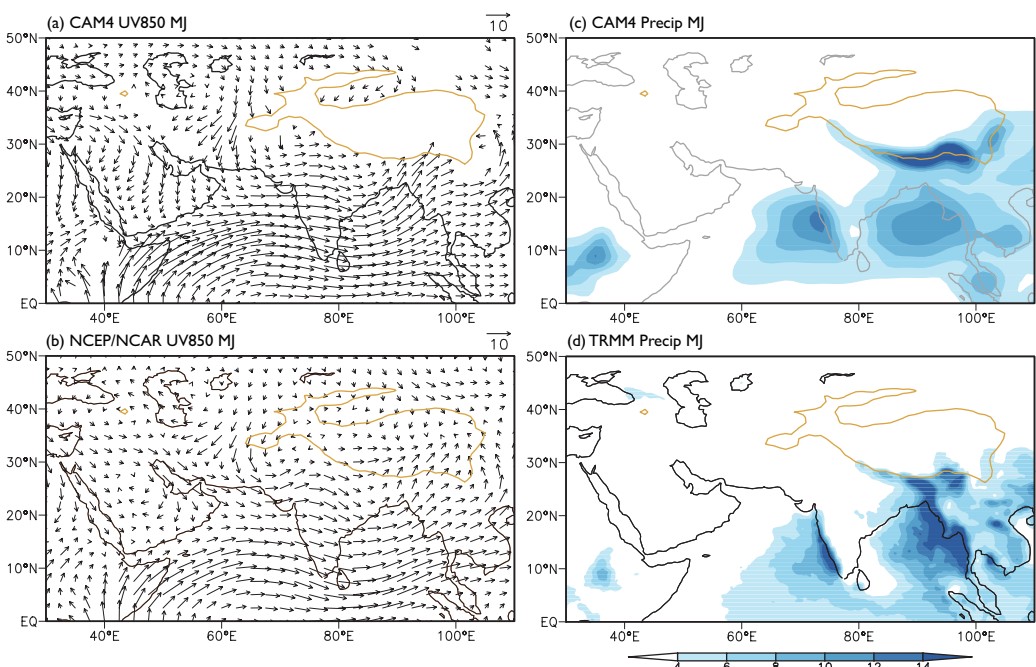

**Figure 3.** Averaged 850 hPa wind vectors (m s$^{-1}$) over Indian monsoon region for May and June in CAM4 (a) and in National Centers for Environmental Prediction/National Center for Atmospheric Research (NCEP/NCAR) reanalysis data (b); and precipitation rates (mm day$^{-1}$) for May and June in CAM4 (c) and in Tropical Rainfall Measuring Mission (TRMM)-retrieved data (d). Yellow line shows the profile of Tibetan Plateau above 2500 m





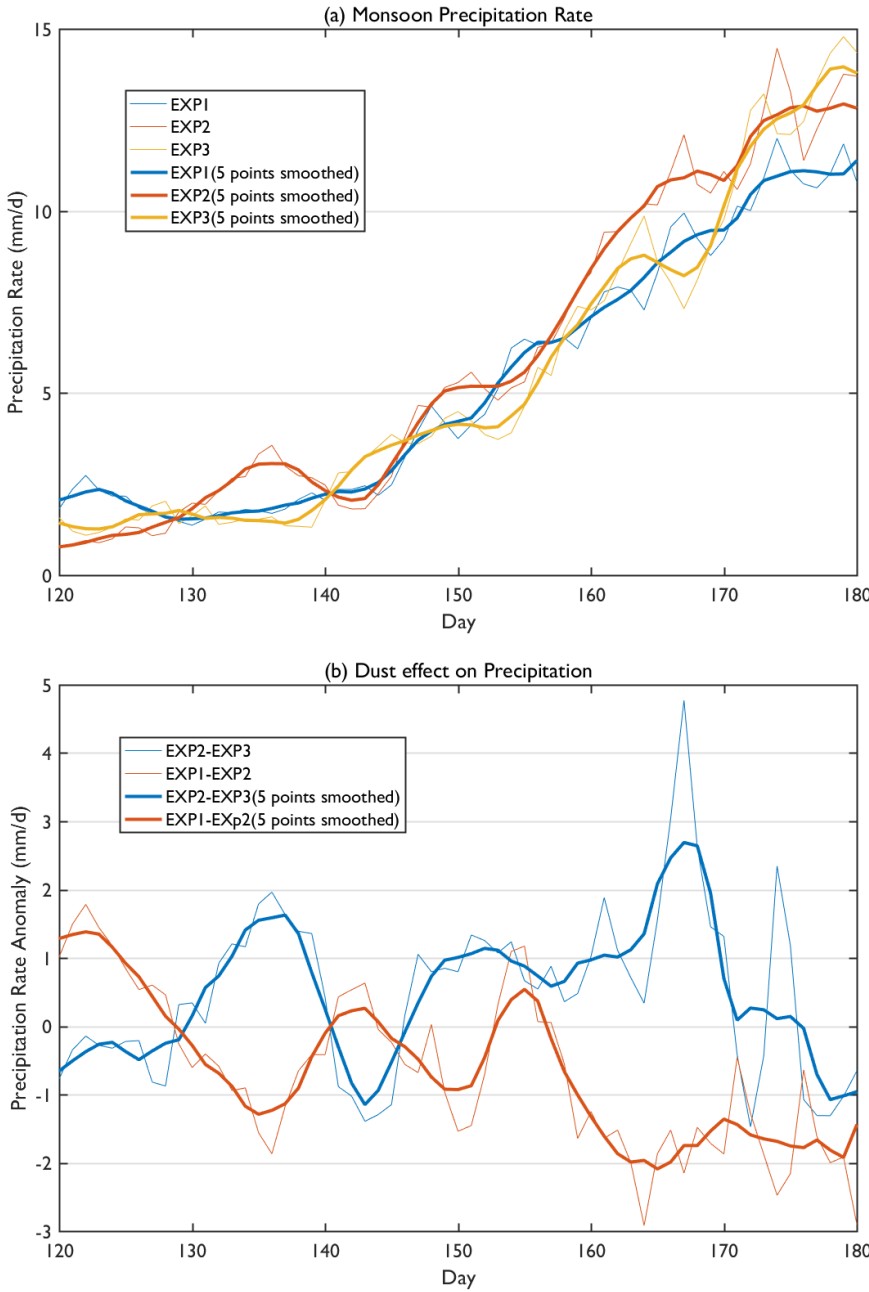

**Figure 4.** Daily precipitation rates (mm day$^{-1}$) during May and June (120-180$^{th}$ days) in three experiments (a) and the differences (mm day$^{-1}$) induced by snow-darkening effect and direct radiative effect (b). Thin lines show the daily values and thick ones are 5-day smoothed. In b, red lines donate snow-darkening effect and blue lines donate direct radiative effect.





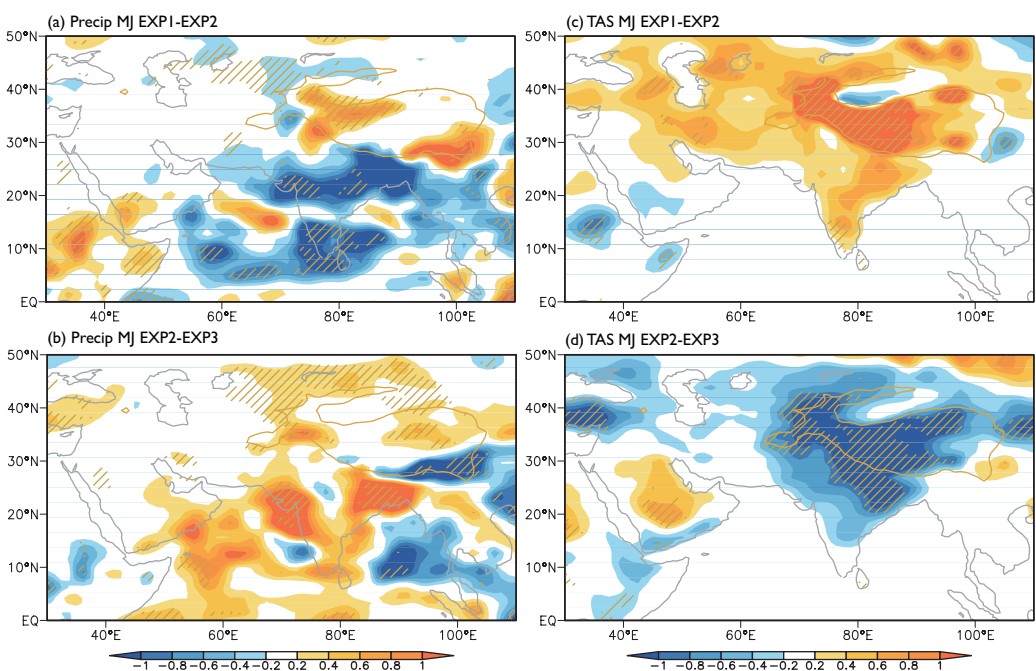

**Figure 5.** Spatial distribution of changes in precipitation rates (a, b, mm day$^{-1}$) and surface temperature (c, d, °C) in May and June induced by snow-darkening effect (top) and direct radiative effect (bottom), respectively. Oblique lines indicate differences significant at 95% confidence level. Yellow line shows the profile of Tibetan Plateau above 2500 m.





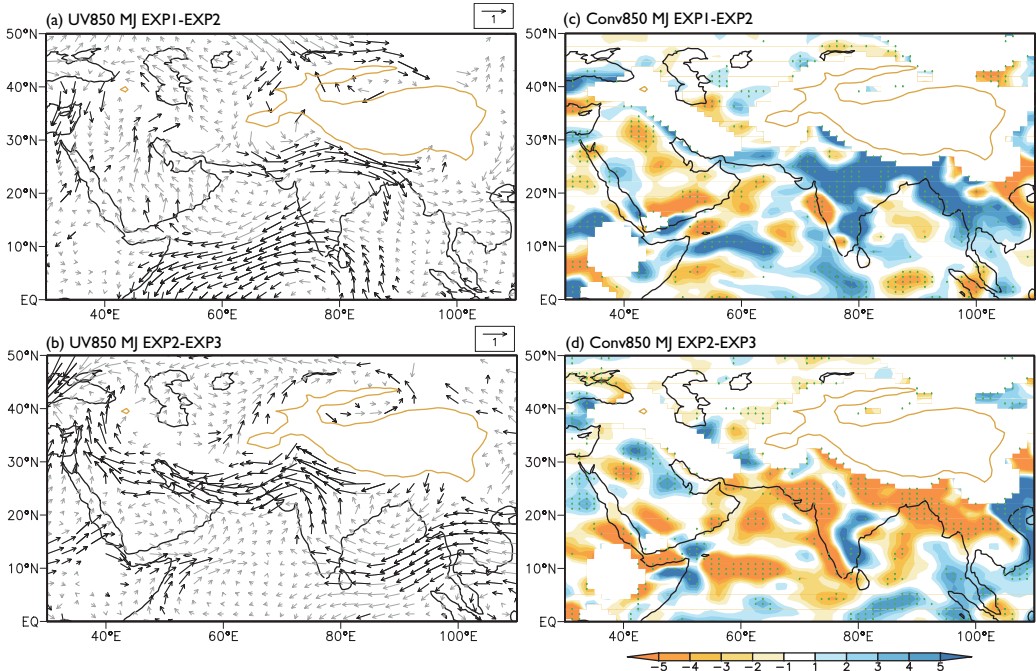

**Figure 6.** Spatial distribution of changes in 850 hPa wind vectors (a, b, m s$^{-1}$) and moisture convergence (c, d, g s kg$^{-1}$ m$^{-1}$) in May and June induced by snow-darkening effect (top) and direct radiative effect (bottom), respectively. Positive values in c and d means divergence anomaly and negative means convergence. Black arrows and green dots indicate differences significant at 90% confidence level. Yellow line shows the profile of Tibetan Plateau above 2500 m.




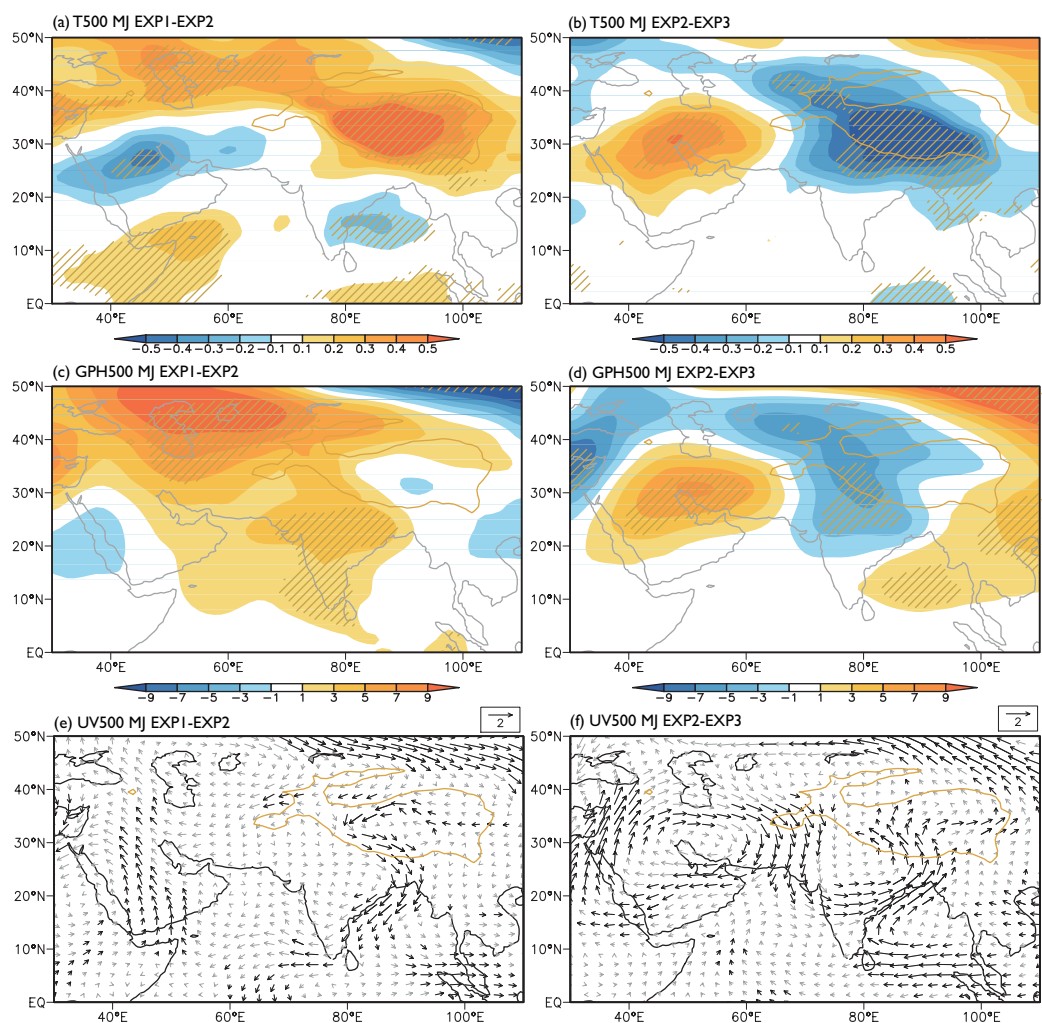

**Figure 7.** Spatial distribution of changes in 500 hPa temperature (a, b, °C), geopotential height (c, d, gpm) and wind vectors (e, f, m s$^{-1}$) in May and June induced by snow-darkening effect (left) and direct radiative effect (right), respectively. Oblique lines and black arrows indicate differences significant at 90% confidence level. Yellow line shows the profile of Tibetan Plateau above 2500 m.




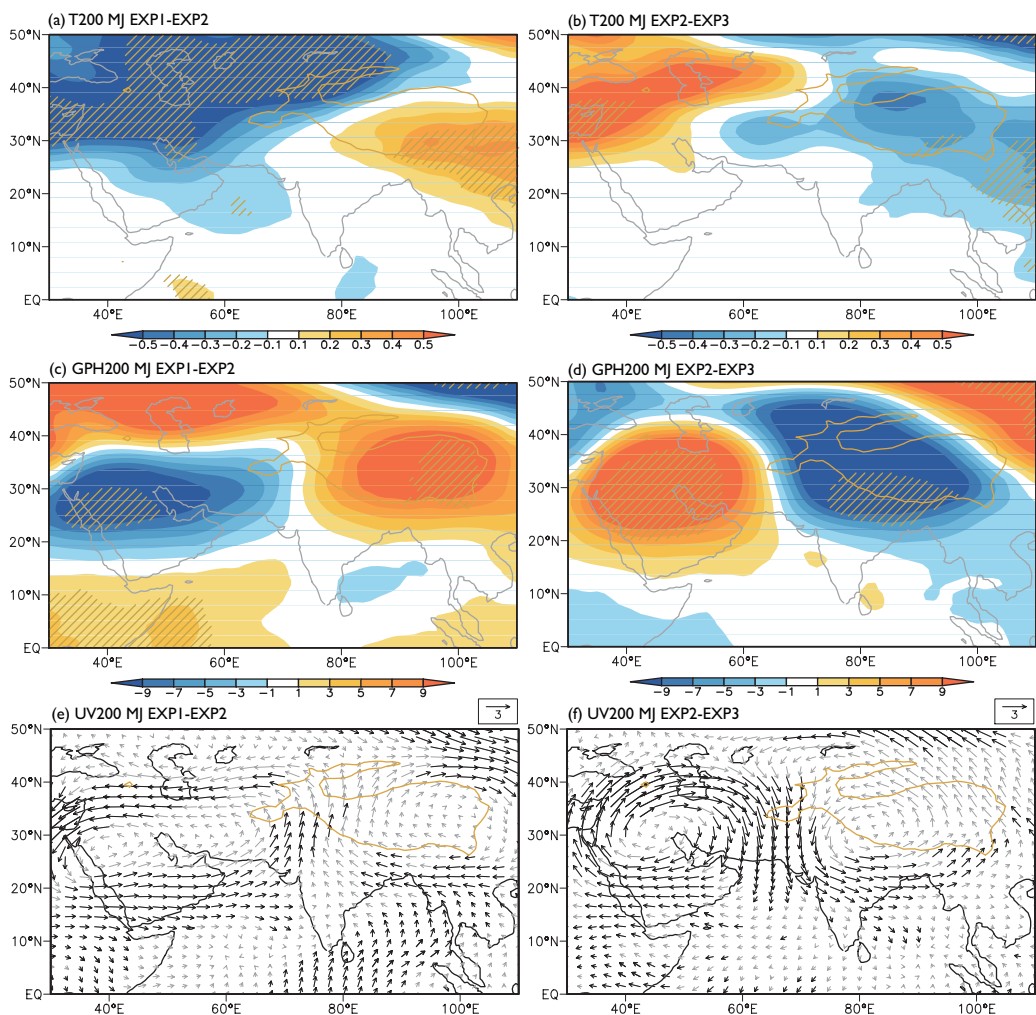

**Figure 8.** Similar with Figure 7, but for 200hPa isobaric level.





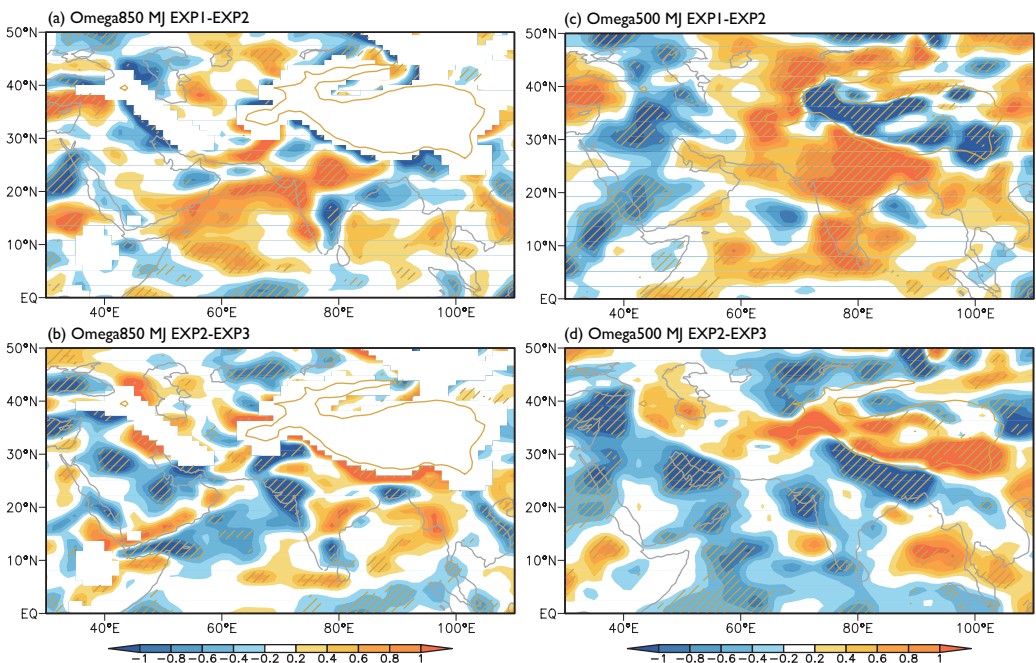

**Figure 9.** Spatial distribution of changes in 850 hPa (a, b) and 500 hPa (c, d) vertical wind speed ($\times 100$, Pa s$^{-1}$) in May and June induced by snow-darkening effect (top) and direct radiative effect (bottom), respectively. Negative values indicate upward flow and positive indicate downward flow. Oblique lines indicate differences significant at 95% confidence level. Yellow line shows the profile of Tibetan Plateau above 2500 m.





**Figure 10.** Changes in meridional (m s$^{-1}$) versus vertical winds ($\times 100$, Pa s$^{-1}$) in a cross section averaged for $80°-95°$E in May and June induced by snow-darkening effect (a) and direct radiative effect (b), respectively. Black arrows indicate differences significant at 95% confidence level and grey ones not statistically-significant.





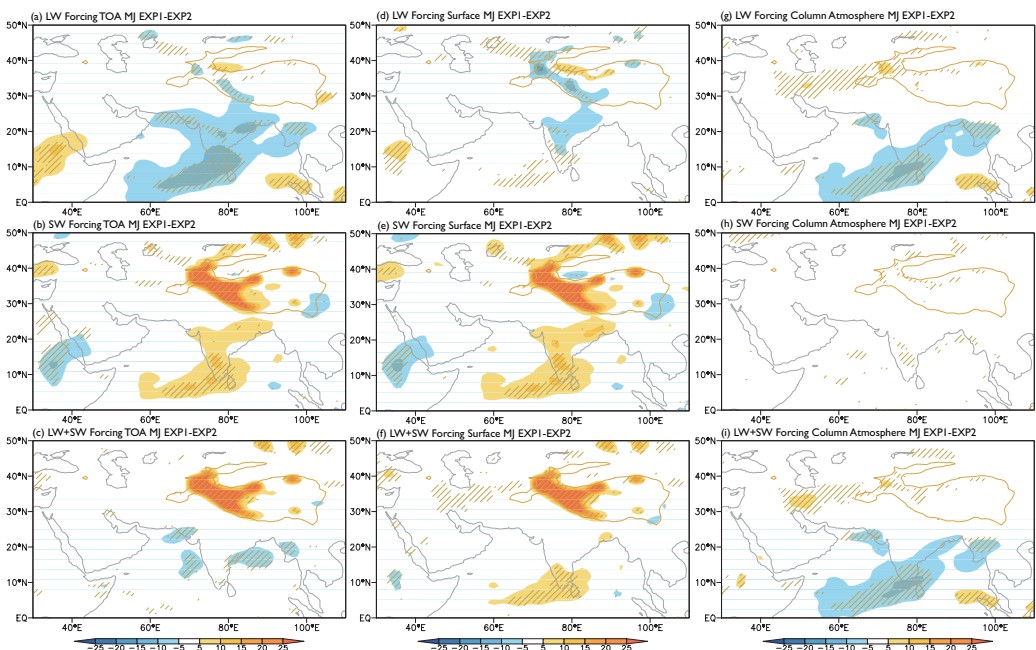

**Figure 11.** .Changes in longwave (top, W m$^{-2}$), shortwave (middle, W m$^{-2}$) and net (longwave+shortwave, bottom, W m$^{-2}$) radiative fluxes during May and June by snow-darkening effect of dust for the top of atmosphere (TOA, a-c), the surface (d-f) and the column atmosphere (g-i). Oblique lines indicate differences significant at 95% confidence level. Yellow line shows the profile of Tibetan Plateau above 2500 m.





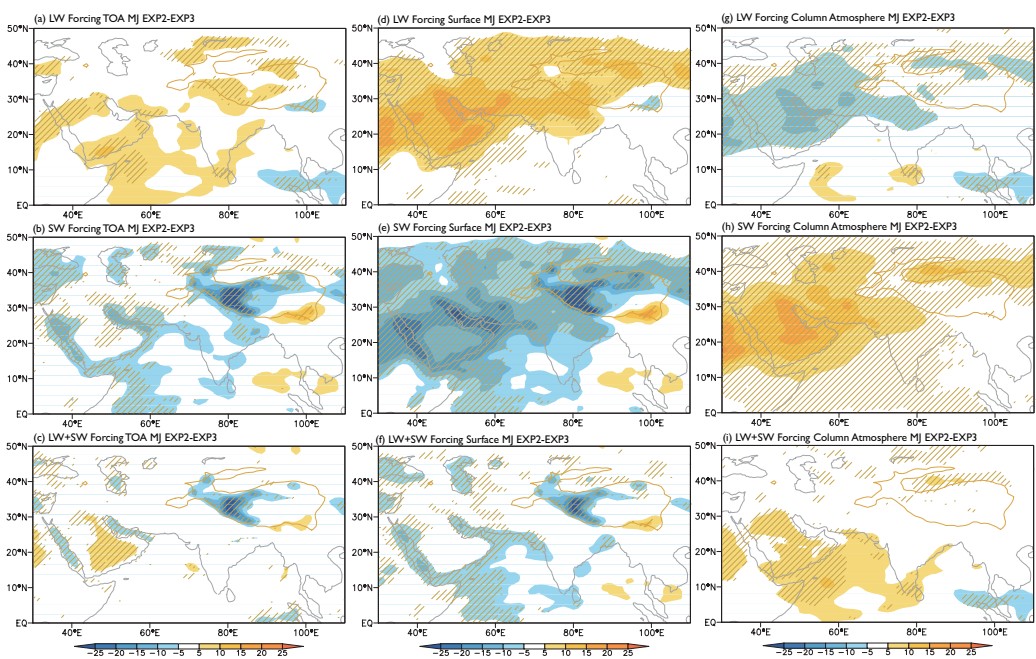

**Figure 12.** Similar with Figure 11, but for direct radiative effect of dust.



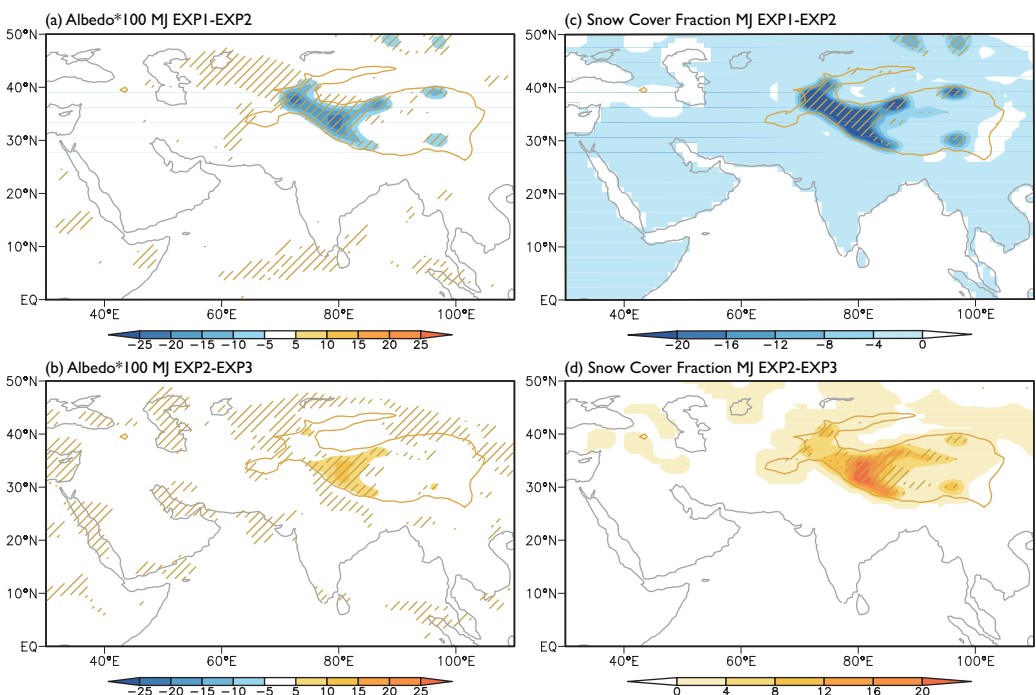

**Figure 13.** Changes in surface albedo (a, b, ×100) and snow cover fractions (c, d, %) over Asia during May and June induced by snow-darkening effect (top) and direct radiative effect (bottom). Oblique lines indicate differences significant at 95% confidence level. Yellow line shows the profile of Tibetan Plateau above 2500 m.



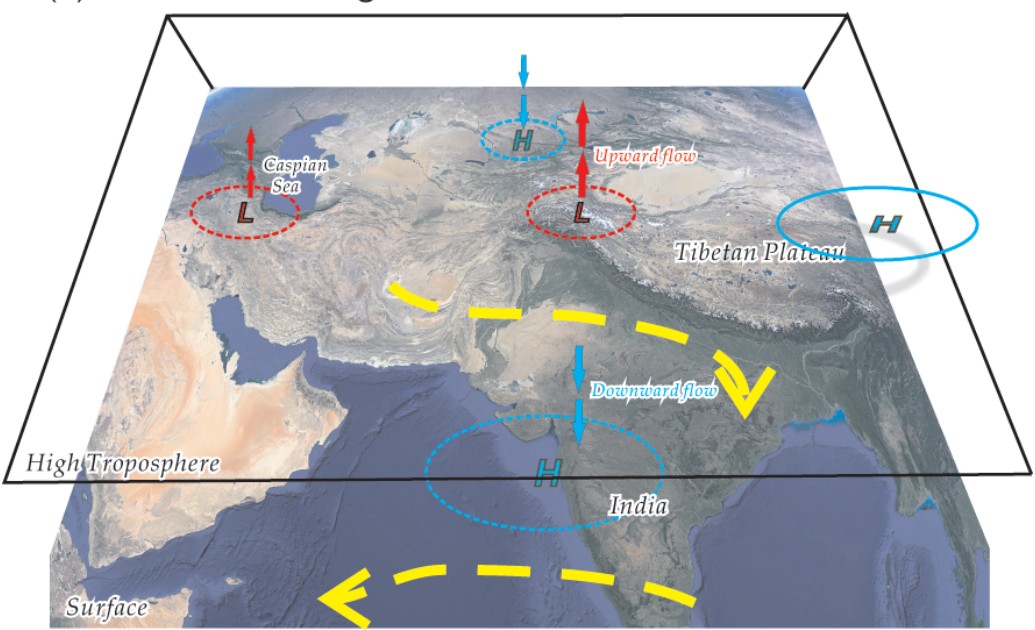

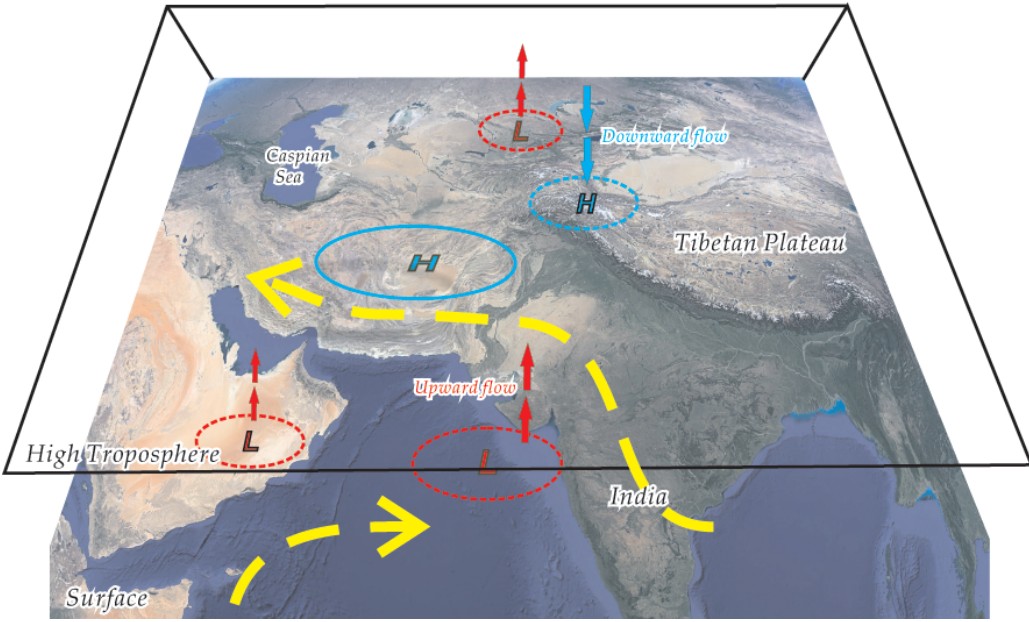

**Figure 14.** Schematic diagrams showing the forcing mechanisms of snow-darkening effect (a) and direct radiative effect (b) of mineral dust on Indian monsoon during the onset. The circles and abbreviations in them denote the anomalous pressure centers: high pressure (blue), low pressure (red), near surface (dashed) and high troposphere (solid). The red and blue arrows indicate the upward and downward air flows, respectively, and the yellow ones present the differences in horizontal winds.