# Peer review of "Snow-darkening versus direct radiative effects of mineral dust aerosol on the Indian summer monsoon onset: role of temperature change over dust sources"

_Atmospheric Chemistry and Physics, 2018_

## Referee Comment (RC1) · Anonymous Referee #1 · 6 Jul 2018

**Review Comments for "Snow-darkening versus direct radiative effects of mineral dust aerosol on the Indian summer monsoon: role of the Tibetan Plateau" by Shi et al.**

The authors conducted a set of GCM simulations to quantify dust SDE and DRE over the Tibetan Plateau and its impacts on Indian monsoon onset. They found that dust SDE and DRE exert opposite effects on Indian monsoon onset and proposed a possible mechanism. The results are interesting and the authors did a generally good job in writing the manuscript. However, some parts of the manuscript still need to be improved, particularly for model descriptions and evaluations. Please see my following comments.

**Major Comments:**
1. Section 2 (Model and Experiments): Since this work is a modeling study, the model descriptions require more details. Here are some examples. (1) What is the new dust size distribution used in CAM4-BAM? How many size bins are used and what are the values for these bins? (2) For dust optical properties, what have been updated? (3) The model simulations did not include aerosol indirect effect but used prescribed CCN. Does this mean that aerosol wet removal through in-cloud process is not included? If so, this could cause large uncertainty in simulations. Please clarify how the prescribed aerosol in-cloud process would affect aerosol wet deposition. (4) The authors used the SNICAR model to deal with snow darkening processes. How does the model handle aerosol-snow interactions? To my understanding, SNICAR assumes external mixing between aerosols and spherical snow grains. But recent studies have suggested that aerosol-snow internal mixing and nonspherical snow shape could significantly affect aerosol-induced snow albedo effects (e.g., Flanner et al., 2012; Liou et al., 2014; Räisänen et al., 2017; He et al., 2018), which may introduce some uncertainty in the simulations here. (5) How does the model deal with the aerosol removal in snowpack? Does it assume a fixed removal efficiency? (6) The way to calculate SDE and DRE by computing the difference between EXP1 and EXP2 and between EXP2 and EXP3 has an underlying assumption that SDE and DRE are linearly additive. However, SDE and DRE could have interactive and nonlinear effects, which makes the calculations above inaccurate. For example, if we refer EXP4 to a new experiment with only SDE enabled, then how different would the result be if calculating DRE by taking the difference between EXP1 and EXP4, compared with "EXP2 minus EXP3". And how different would the result be if calculating SDE by taking the difference between EXP4 and EXP3, compared with "EXP1 minus EXP2". Do the authors have any suggestions on which way of calculation is more accurate in terms of quantifying dust SDE and DRE?
References:
Flanner, M. G., et al.: Enhanced solar energy absorption by internally-mixed black carbon in snow grains, Atmos. Chem. Phys., 12(10), 4699–4721, 2012.
He, C., et al.: Impact of grain shape and multiple black carbon internal mixing on snow albedo: Parameterization and radiative effect analysis, J. Geophys. Res.-Atmos., 123, 1253–1268, 2018.
Liou, K. N., et al.: Stochastic parameterization for light absorption by internally mixed BC/dust in snow grains for application to climate models, J. Geophys. Res.-Atmos., 119, 7616–7632, 2014.
Räisänen, P., et al.: Effects of snow grain shape on climate simulations: Sensitivity tests with the Norwegian Earth System Model, The Cryosphere, 2017.

2. Section 3.1 (Model validation): (1) For the AOD evaluation, since the model simulation did not include non-dust aerosols, it is not an apple-to-apple comparison for modeled and MISR AOD here. The AOD comparison did not give us very useful information. If the authors want to use

total AOD from observations, the model simulations need to include all aerosol types. If the authors only want dust AOD, maybe CALIPSO observations could help. Focusing on AOD over dust source regions can also be a way to evaluate modeled dust AOD, but in that case it is difficult to know how the model performs in terms of dust transport, particularly over remote regions such as the Tibetan Plateau. Besides, even over the dust source regions such as north of Tibetan Plateau, the modeled AOD is much smaller than MISR AOD. What would be the possible reasons? (2) Also it seems that MODIS AOD is better than MISR AOD at least over dust source regions, due to the MODIS deep blue retrieval algorithm. Why did the authors select MISR instead of MODIS? (3) The authors described in detail the consistency and inconsistency between model simulations and observations in terms of AOD, snow cover, and monsoon climatology, but it appears that not enough explanations have been provided for the model-observation differences. The readers may also want to know the reasons causing the model-observation discrepancies, which would be very useful for future model improvements. (4) Since the snow darkening effect (i.e., albedo reduction) is one focus in this work, it would be straightforward to consider evaluating modeled snow/surface albedo at least over the Tibetan Plateau, for example, by comparing with MODIS albedo product. Is there any specific reason for the authors to leave out this part?

**Minor Comments:**

1. Page 1, Line 16: I suggest replacing "clarified" with "quantified".

2. Page 2, Line 10: Please remove "reflect," since reflection is part of scattering.

3. Page 2, Lines 31-34: For the authors' information, some recent studies on BC/dust SDE are missing here, which improved the understanding of aerosol SDE particularly over the Tibetan Plateau. Some examples are listed as follows.
References:
He, C., et al: Black carbon radiative forcing over the Tibetan Plateau, Geophys. Res. Lett., 41, 7806–7813, 2014.
Zhao, C., et al.: Simulating black carbon and dust and their radiative forcing in seasonal snow: a case study over North China with field campaign measurements, Atmos. Chem. Phys., 14, 11475-11491, 2014.
Lee, W.-L., et al.: Impact of absorbing aerosol deposition on snow albedo reduction over the southern Tibetan plateau based on satellite observations, Theor. Appl. Climatol., 129(3-4), 1373-1382, 2017.
Niu, H.W., et al.: Distribution of light-absorbing impurities in snow of glacier on Mt. Yulong, southeastern Tibetan Plateau, Atmos. Res., 197, 474-484, 2017.

4. Section 1 (Introduction): It seems that the authors did not mention their motivation to focus on the Tibetan region particularly. Thus, I suggest adding a short paragraph to highlight the importance of Tibetan Plateau (such as its role in altering Asian water resources and hydrological cycle), although the authors already mentioned a little bit in the descriptions of dust effects.

5. Page 9, Line 6: please remove "is" before "occurs".

---

## Referee Comment (RC2) · Anonymous Referee #2 · 8 Aug 2018

**Review of "Snow-darkening versus direct radiative effects of mineral dust aerosol on the Indian summer monsoon: role of the Tibetan Plateau" by Shi et al.**

This paper examines the dust snow-darkening (SDE) and direct radiative effects (DRE) on Indian summer monsoon (ISM) with global climate model simulations. The authors found that dust SDE (DRE) tends to induce a warming (cooling) over Tibetan Plateau, and weakens (intensifies) the ISM. The main findings of this manuscript are contradictory to previous studies, but the authors did not provide convincing explanations. Thus, this manuscript needs careful revisions to meet the standard of Atmospheric Chemistry and Physics and resubmitted.

**Major comments:**

The Indian summer monsoon is primarily driven by the thermal contrast between land and ocean (Wang et al., 2000). The up troposphere meridional temperature gradient south of Tibetan Plateau is one of the key controls of the Indian summer monsoon (Li and Yanai, 1996). An up troposphere warming over TP tends to increase the meridional temperature gradient, and intensifies the Indian summer monsoon (Wu et al., 2005; Liu et al., 2001). Previous studies show that both DRE and SDE of absorbing aerosols could induce a warming around TP and intensify the Indian summer monsoon (EHP effect), which is in general consistent with the observed relationship. In this study, however, the authors found the Indian summer monsoon is intensified (weakened) associated with cooling (warming) over TP, which is just opposite to previous studies. The authors should carefully check the model settings and give some explanations.

The authors found that the dust SDE induced TP warming tends to weakens Indian summer monsoon, which is opposite to what found in Qian et al. 2011. The authors stated that the opposite response is due to the difference in TP warming center distribution. Lau et al. 2010 found that aerosol SDE could produce an "elevated-heat-pump (EHP)" effect and increase the precipitation over Indian in May. Their results is in generally consistent with Qian et al. 2011, although the warming center is over western TP. The authors should provide convincing explanations why the response to dust SDE in this study is contradictory to previous studies.

The dust DRE impacts on Indian summer monsoon is also inconsistent with previous studies, and the results are difficult to understand. The authors found dust DRE could induce a significant cooling over TP, and the cooling is attributed to snow-albedo feedback. The dust AOD is very small over TP (less than 0.05), which implies very weak DRE. How such small DRE produce strong snow-albedo feedback over TP? The feedback processes should be detailed explained. More confusing thing is that the Indian summer monsoon (ISM) is intensified associated with the TP cooling, which is similar to the response induced by TP warming (Lau et al., 2006). The authors simply explained it as a response to downward motion right over TP, which is not convincing. Please provide detailed explanations and supportive reference.

In this study, the CAM4 was run with prescribed climatological SST and sea ice. The SST response to aerosol forcing (slow response) is not taken into account. Many previous works showed that the slow response can play a dominant role in the total response of Indian summer monsoon to aerosol forcing (Ganguly et al., 2012). Many previous studies investigated dust impacts on ISM with coupled simulations (e.g. Qian et al, 2011). It could be a possible reason why the monsoon response is opposite to previous studies. Thus, the authors should run coupled simulations and make a comparison with current results.

This study investigate the dust impacts on Indian summer monsoon. However, only the dust effect during the monsoon onset periods (May and June) is investigated. The Indian summer monsoon is from June to August (or September). Please show the monsoon response in July and August, for the dust concentration is still high in Indian at that time (Gu et al., 2016). The response of ISM could be quite different in July and August, for dust DRE impacts could be more important at that time. Only with an examination of the response in entire monsoon period, the title of this manuscript could be appropriate.

**Other comments:**

Page 1, Line 2:"have" should be "has".

Page 4, Line 6-7:Please give more explanations on "snow-darkening and direct radiative feedbacks". Does it mean the permit of dust snow-darkening and direct radiative effects in simulations? What is the meaning of feedback?

Page 5, Line 13: Please provide references for the two branches of Indian summer monsoon.

Page 5, Line 25: If EXP1-EXP2 equals to the impacts of dust SDE, please use the dust SDE in the rest of manuscript for consistency. So do the cases for EXP2-EXP3.

Page 5, Line 26: Please clarify the definition of "Indian monsoon area".

Page 5, Line 28: Indian summer monsoon lasts from June to August. Please show the precipitation change in July and August, as well.

Page 6, Line 9:Why dust SDE induces significant cooling over Tibetan Plateau? The dust AOD is very small over Tibetan Plateau.

Page 6, Line 15:Why is the southern branch of the monsoon westerly significantly decreased?

Page 8, Line 10: Please show the dust snow forcing (outputted by SNICAR), dust deposition (dry and wet), and dust concentration in snow over TP and their seasonal variation. A comparison with previous studies (e.g., Qian et al., 2011) is also needed.

Page 8, Line 23: Please explain the feedback.

Page 8, Line 23: Dust aerosols could absorb both shortwave and longwave radiative fluxes. Why the longwave radiative flux change is negative?

Page 8, Line 34: The dust AOD is very small over TP (less than 0.05), which implies very weak DRE. How such weak DRE produce significant snow cover increase and surface cooling over TP? It could not be simply attributed to snow-albedo feedback.

Page 9, Line 19-30: In Lau et al. 2010, they found that TP warming tends to increase the Indian precipitation in May, and the warming center is located at western TP. Their result is consistent with Qian et al. 2010, but different with the results of this manuscript. Explanations are needed here.

Page 10, Line 12: How could downward motion right over TP induce an upward motion over Indian? Is it noticed any previous studies? Please provide more explanations as well as the references.

**Figures:**

Figure 2 and Figure 3 could be put in the supplement, for they are too many figures for this manuscript.

Figure 4:Please use the specific date in figure 4 (e.g. May 1st).

Figure 4:Please specify the regions of precipitation change.

Figure 5:Please display the precipitation and surface temperature with different color tables.

Figure 5 and so on: Please show "SDE" and "DRE" in figure title.

Figure 5 to Figure 10: There are too many figures for this part. Decide what is important and put the rest in supplement.

**References:**

Ganguly, D., P. J. Rasch, H. Wang, and J.-h. Yoon (2012), Fast and slow responses of the South Asian monsoon system to anthropogenic aerosols, Geophysical Research Letters, 39(18), L18804, doi:10.1029/2012GL053043.

Gu, Y., Xue, Y., De Sales, F., Liou, K. N.: A GCM investigation of dust aerosol impact on the regional climate of North Africa and South/East Asia, Clim. Dyn., 46, 2353–2370, 2016.

Lau, K. M., Kim, M. K., and Kim, K. M.:. Asian monsoon anomalies induced by aerosol direct forcing: the role of the Tibetan Plateau, Clim. Dyn., 26, 855–664, 2006.

Lau, K.-M., Kim, M. K., Kim, K.-M., and Lee, W. S.: Enhanced surface warming and accelerated snow melt in the Himalayas and Tibetan Plateau induced by absorbing aerosols, Environ. Res. Lett., 5, 025204 doi:10.1088/1748-9326/5/2/025204, 2010.

Li, C., and M. Yanai (1996), The Onset and Interannual Variability of the Asian Summer Monsoon in Relation to Land–Sea Thermal Contrast, Journal of Climate, 9(2), 358-375, doi:10.1175/1520-0442(1996)009<0358:TOAIVO>2.0.CO;2.

Liu, X., and M. Yanai (2001), Relationship between the Indian monsoon rainfall and the tropospheric temperature over the Eurasian continent, Quarterly Journal of the Royal Meteorological Society, 127(573), 909-937, doi:10.1002/qj.49712757311.

Qian, Y., Flanner, M., Leung, L., and Wang, W.: Sensitivity studies on the impacts of Tibetan Plateau snowpack pollution on the Asian hydrological cycle and monsoon climate, Atmos. Chem. Phys., 11(5), 1929–1948, doi: 10.5194/acp-11-1929-2011, 2011.

Wu, B. (2005), Weakening of Indian summer monsoon in recent decades, Advances in Atmospheric Sciences, 22(1), 21-29, doi:10.1007/BF02930866.

Wang, B., R. Wu, and K.-M. Lau (2001), Interannual Variability of the Asian Summer Monsoon: Contrasts between the Indian and the Western North Pacific–East Asian Monsoons, Journal of Climate, 14(20), 4073-4090, doi:10.1175/1520-0442(2001)014<4073:Ivotas>2.0.Co;2.

---

## Author Comment (AC1) · 11 Sep 2018

**Review Comments for "Snow-darkening versus direct radiative effects of mineral dust aerosol on the Indian summer monsoon: role of the Tibetan Plateau" by Shi et al.**

The authors conducted a set of GCM simulations to quantify dust SDE and DRE over the Tibetan Plateau and its impacts on Indian monsoon onset. They found that dust SDE and DRE exert opposite effects on Indian monsoon onset and proposed a possible mechanism. The results are interesting and the authors did a generally good job in writing the manuscript. However, some parts of the manuscript still need to be improved, particularly for model descriptions and evaluations. Please see my following comments.

RE: Thanks for the positive comments.

Major Comments:
1. Section 2 (Model and Experiments): Since this work is a modeling study, the model descriptions require more details. Here are some examples. (1) What is the new dust size distribution used in CAM4-BAM? How many size bins are used and what are the values for these bins?

RE: There are four size bins (0-1.0 µm; 1.0-2.5 µm; 2.5-5.0 µm; 5.0-10.0 µm) in diameters used in this study. In this version, the precentages of emission (0.02, 0.09, 0.27, 0.62) is modified to allow more large particles.

We noted this in the revision (Page 3 Lines 28-32): "The dust cycle including the emission, transport and deposition, is parameterized in CAM4 and its radiative feedbacks are also calculated on line. The dust sizes in CAM4 contain four bins of 0-1.0 µm; 1.0-2.5 µm; 2.5-5.0 µm; 5.0-10.0 µm in diameters, respectively (Mahowald et al., 2006). The CAM4-BAM has been improved by an optimized soil erodibility map and a new size distribution for dust emission (the percentages for four bins are 0.02,0.09,0.27,0.62, respectively), as well as updated optical properties for radiation budget, to present a better performance on simulating the global dust cycle (Albani et al., 2014)"

(2) For dust optical properties, what have been updated?

RE: Several optics in the CAM4-BAM were improved, as introduced in Page 7 of Albani et al., (2014). Due to limited space, we did not introduce these changes in our paper. We referred to Albani et al.' paper for detailed information (Page 3 Lines 31-32).

(3) The model simulations did not include aerosol indirect effect but used prescribed CCN. Does this mean that aerosol wet removal through in-cloud process is not included? If so, this could cause large uncertainty in simulations. Please clarify how

the prescribed aerosol in-cloud process would affect aerosol wet deposition.

RE: The reason we chose the model CAM4 not CAM5 is that CAM4 does not include aerosol indirect effect, which is complicated with large uncertainty and not our focus. Certainly, the choice of CAM4 leads to other uncertainty, e.g., the in cloud removal of dust, as the reviewer pointed out. We admit this bias in simulations, however, the bias in wet deposition does not affect our discussion because wet deposition occupies a small part of total deposition over dust source regions.

We emphasized this possible bias in the revision (Page 3 Lines 32-33; Page 4 Lines 1-2): "In CAM4-BAM, the SDE of all aerosols are enabled but the indirect effect is not considered, which means that the aerosol changes in cloud process as condensation nuclei are prescribed. Wet removal through in-cloud process is not considered, which may induce bias of dust deposition on snow over Asia. "

(4) The authors used the SNICAR model to deal with snow darkening processes. How does the model handle aerosol-snow interactions? To my understanding, SNICAR assumes external mixing between aerosols and spherical snow grains. But recent studies have suggested that aerosol-snow internal mixing and nonspherical snow shape could significantly affect aerosol-induced snow albedo effects (e.g., Flanner et al., 2012; Liou et al., 2014; Räisänen et al., 2017; He et al., 2018), which may introduce some uncertainty in the simulations here.

RE: Thanks for the comment. SCINAR assumes external mixing between aerosols and spherical snow grains, which may induce uncertainty.

We referred the publications and discussed the uncertainty of simplification of spherical snow grains in this model (Page 4 Lines 7-9): "Of note is that SCINAR assumes external mixing between aerosols and spherical snow grains, however, aerosol-snow internal mixing and nonspherical snow shape could significantly affect aerosol-induced snow albedo effects, based on recent studies (Flanner et al., 2012; Liou et al., 2014; Räisänen et al., 2017; He et al., 2018)."

(5) How does the model deal with the aerosol removal in snowpack? Does it assume a fixed removal efficiency?

RE: Yes. In the model, it assume a fixed removal efficiency, that is, the removal by meltwater is proportional to its mass mixing ration (multiplied by a scavenging factor).

(6) The way to calculate SDE and DRE by computing the difference between EXP1 and EXP2 and between EXP2 and EXP3 has an underlying assumption that SDE and DRE are linearly additive. However, SDE and DRE could have interactive and nonlinear effects, which makes the calculations above inaccurate. For example, if we

refer EXP4 to a new experiment with only SDE enabled, then how different would the result be if calculating DRE by taking the difference between EXP1 and EXP4, compared with "EXP2 minus EXP3". And how different would the result be if calculating SDE by taking the difference between EXP4 and EXP3, compared with "EXP1 minus EXP2". Do the authors have any suggestions on which way of calculation is more accurate in terms of quantifying dust SDE and DRE?

RE: Yes, the reviewer is right. The nonlinear term indeed exists because of the interaction between two effects. A good way to examine the nonlinear term is to conduct a fourth experiment with only SDE enabled and see whether EXP1-EXP4 and EXP2 and EXP3 is consistent (if yes, it means the nonlinear term can be neglected). In this study, we mainly focused the SDE of dust so we used EXP1-EXP2 (not EXP4-EXP3) because we considered the EXP1 as control experiment and in real world these two effects are indeed enabled. However, due to limited time for final response phase, it is difficult to finish the EXP4 to examine the nonlinear term (We have already added three more experiments on black carbon. Please see the response for RC2). If the reviewer insists, we wish to finish the EXP4 in next phase.

References:
Flanner, M. G., et al.: Enhanced solar energy absorption by internally-mixed black carbon in snow grains, Atmos. Chem. Phys., 12(10), 4699–4721, 2012.
He, C., et al.: Impact of grain shape and multiple black carbon internal mixing on snow albedo: Parameterization and radiative effect analysis, J. Geophys. Res.-Atmos., 123, 1253–1268, 2018.
Liou, K. N., et al.: Stochastic parameterization for light absorption by internally mixed BC/dust in snow grains for application to climate models, J. Geophys. Res.-Atmos., 119, 7616–7632, 2014.
Räisänen, P., et al.: Effects of snow grain shape on climate simulations: Sensitivity tests with the Norwegian Earth System Model, The Cryosphere, 2017.

2. Section 3.1 (Model validation): (1) For the AOD evaluation, since the model simulation did not include non-dust aerosols, it is not an apple-to-apple comparison for modeled and MISR AOD here. The AOD comparison did not give us very useful information. If the authors want to use total AOD from observations, the model simulations need to include all aerosol types. If the authors only want dust AOD, maybe CALIPSO observations could help. Focusing on AOD over dust source regions can also be a way to evaluate modeled dust AOD, but in that case it is difficult to know how the model performs in terms of dust transport, particularly over remote regions such as the Tibetan Plateau. Besides, even over the dust source regions such as north of Tibetan Plateau, the modeled AOD is much smaller than MISR AOD. What would be the possible reasons?

RE: Thanks. We used the CALIPSO data instead in the revision. Compared to the dust

AOD in CALIPSO data, the simulated AOD is smaller, especially over the source areas (Figure R1). The possible reasons are as follows. Over the dust source, there are lots of dust with larger particle sizes but in the model, the considered dust particles are restricted to less than 10μm. Thus, the dust forcing is underestimated due to less coarser dusts in the current global climate models (Kok et al., 2017). Also, dust model may have quite large differences in simulating vertical distribution, emission, deposition, and surface concentration of dust (Pu and Ginoux, 2018), which affects the AOD as an integrated variable. Furthermore, the spatial resolution of model is not fine enough, which fails to well resolve the complex topography and dust sources over East Asia.

We mentioned these in the revision (Page 5 Lines 5-8): "The simulated absolute values of dust AOD over Arabian Peninsular, southwestern slope of the TP and Taklimakan desert are biased low because the considered dust particles are restricted to less than 10.0 μm and the dust forcing is underestimated due to less coarser dusts in the current global climate models (Kok et al., 2017). "

[Figure]

Figure R1: Averaged dust aerosol optical depth over Asia for May and June in CAM4 (a) and in Cloud-Aerosol Lidar and Infrared Pathfinder Satellite Observation (CALIPSO)-retrieved data for 2007-2011 (b); and dust deposition flux including both dry and wet deposition for March and April (c) and for May and June (d).

(2) Also it seems that MODIS AOD is better than MISR AOD at least over dust source regions, due to the MODIS deep blue retrieval algorithm. Why did the authors select MISR instead of MODIS?

RE: See the response above. We used CALIPSO data for dust because it can be directly compared to our simulated AOD.

(3) The authors described in detail the consistency and inconsistency between model

simulations and observations in terms of AOD, snow cover, and monsoon climatology, but it appears that not enough explanations have been provided for the model observation differences. The readers may also want to know the reasons causing the model observation discrepancies, which would be very useful for future model improvements.

RE: For the reasons for AOD differences, we discussed it in previous response 2-(1). The main cause for the underestimation on snow cover fraction and precipitation over monsoon regions is obviously the coarser resolution in our model. For example, our model can not resolve the high mountains over Tianshan mountains and western Tibetan Plateau, wihch the simulated snow cover fraction is relative smaller. Similarly, thin high topography over western edges of Indian subcontinent and Indo-China Peninsula is also partly missed so the observed maximal precipitation at these regions are also smaller in model.

In the revision, we added these explanations (Page 5 Lines 5-8, 17-20): "The simulated absolute values of dust AOD over Arabian Peninsular, southwestern slope of the TP and Taklimakan desert are biased low because the considered dust particles are restricted to less than 10.0 µm and the dust forcing is underestimated due to less coarser dusts in the current global climate models (Kok et al., 2017).", "Over the western TP, the MODIS observation presents a fraction larger than 80% but the simulated fraction is smaller. In particular, the model underestimates the elevations of finer-scale mountains and corresponding snow cover fractions due to the coarser resolution, e.g., over the Tianshan mountains. "

(4) Since the snow darkening effect (i.e., albedo reduction) is one focus in this work, it would be straightforward to consider evaluating modeled snow/surface albedo at least over the Tibetan Plateau, for example, by comparing with MODIS albedo product. Is there any specific reason for the authors to leave out this part?

RE: Thanks. We did not show the albedo because the albedo is directly controlled by the snow cover fraction. Compared to the MODIS data over the TP (Meng et al., 2018), the model captures its spatial distribution but overestimates the surface albedo, which is similar with multi-model ensembles (Li et al., 2016), mainly due to the overestimated snow cover fractions.

Minor Comments:
1. Page 1, Line 16: I suggest replacing "clarified" with "quantified".

RE: We kept unchanged because we can not quantify their links.

2. Page 2, Line 10: Please remove "reflect," since reflection is part of scattering.

RE: Removed.

3. Page 2, Lines 31-34: For the authors' information, some recent studies on BC/dust SDE are missing here, which improved the understanding of aerosol SDE particularly over the Tibetan Plateau. Some examples are listed as follows.
References:
He, C., et al: Black carbon radiative forcing over the Tibetan Plateau, Geophys. Res. Lett., 41, 7806– 7813, 2014.
Zhao, C., et al.: Simulating black carbon and dust and their radiative forcing in seasonal snow: a case study over North China with field campaign measurements, Atmos. Chem. Phys., 14, 11475-11491, 2014.
Lee, W.-L., et al.: Impact of absorbing aerosol deposition on snow albedo reduction over the southern Tibetan plateau based on satellite observations, Theor. Appl. Climatol., 129(3-4), 1373-1382, 2017.
Niu, H.W., et al.: Distribution of light-absorbing impurities in snow of glacier on Mt. Yulong, southeastern Tibetan Plateau, Atmos. Res., 197, 474-484, 2017.

RE: Thanks for the references and we added them (Page 3 Line 4).

4.Section 1 (Introduction): It seems that the authors did not mention their motivation to focus on the Tibetan region particularly. Thus, I suggest adding a short paragraph to highlight the importance of Tibetan Plateau (such as its role in altering Asian water resources and hydrological cycle), although the authors already mentioned a little bit in the descriptions of dust effects.

RE: Following the comments of another reviewer, we largely reorganized the results and discussion in the revision. In the original manucript, the emphasized role of Tibetan Plateau may be not approprite (the reasons can be detailed introduced in response RC2). Thus, we do not add a paragraph to introduce the role of TP.

5. Page 9, Line 6: please remove "is" before "occurs".

RE: Removed.

---

## Author Comment (AC2) · 11 Sep 2018

**Review of "Snow-darkening versus direct radiative effects of mineral dust aerosol on the Indian summer monsoon: role of the Tibetan Plateau" by Shi et al.**

This paper examines the dust snow-darkening (SDE) and direct radiative effects (DRE) on Indian summer monsoon (ISM) with global climate model simulations. The authors found that dust SDE (DRE) tends to induce a warming (cooling) over Tibetan Plateau, and weakens (intensifies) the ISM. The main findings of this manuscript are contradictory to previous studies, but the authors did not provide convincing explanations. Thus, this manuscript needs careful revisions to meet the standard of Atmospheric Chemistry and Physics and resubmitted.

RE: Thanks for the comments. The reviewer said that the main findings of our paper are contradictory to previous studies. But we do not agree with the reviewer's viewpoint although we admit that some inaccurate arguments in the original manuscript may mislead the readers. We overemphasized the role of Tibetan Plateau in the original manuscript and it is actually not accurate based on new experiments and results.

We agree with the reviewer that we did not provide convincing explanations previously, thus, we conducted three additional experiments, with a special focus on black carbon. The aim is to examine whether the SDE and DRE of black carbon is similar with mineral dust or not. In most previous studies, black carbon is mainly considered to understand the effect of absorbing aerosols but mineral dust is indeed different from black carbon both for spatial distribution and radiative effect. Fortunately, the black carbon experiments tell that the SDE and DRE both intensify the Indian summer monsoon during the onset, which are in good agreements with previous studies (e.g., Lau et al., 2006; Qian et al., 2011).

The SDE of black carbon warms the surface over Tibetan Plateau (TP) and intensifies the monsoon during the onset, consistent with what were found in Qian et al (2011). Interestingly, the same model with the SDE of dust gives a quite different response of monsoon, which indicates different mechanisms behind SDE of dust and black carbon. As we proposed, the spatial distributions of dust and black carbon are not similar. The main difference is that black carbon from the industrial countries is generally transport eastwards and scarcely into upwind Central Asia. Central Asia is also covered by snow although far less than TP. As a result, the forcing of black carbon is restricted to TP but the forcing of dust is over Central Asia and TP. Westwards/northwestwards expansion of warming also shifts the pattern of low level circulation change, which weakens the summer monsoon.

The DRE of black carbon also warms the surface over the TP and intensifies the monsoon during the onset, consistent with Lau et al. (2006). Although the 3D distribution of black carbon are not the same, our experiments also support the DRE of black carbon can strengthen the monsoon. Comparing the responses to dust and black carbon under the same model and experiment design, we found that the summer monsoon during the onset is both intensified no matter whether it is warming or cooling over the TP. As the reviewer said, the TP cooling is unlikely to intensify the

monsoon (at least few evidence support it). We agree it and ascribe the monsoon strengthening to the warming over Arabian Peninsula/Middle East, which also gains strong support from previous researches (Vinoj et al., 2014; Jin et al., 2014; Solomon et al., 2015). Although the TP cooling tends to weaken the monsoon, the Middle East warming overacts and induces a stronger monsoon instead. In addition, the simulated TP cooling might be model dependent because DRE of dust is largely uncertain and depends closely on the size distributions, optical properties and etc (Kok et al., 2017). Anyway, our results support the important role of Middle East warming.

To summarize, we found different mechanisms for Indian monsoon to dust and black carbon forcing. The significant contributions from temperature changes over source areas (Central Asia for SDE and Middle East for DRE, respectively) are highlighted. The role of TP we proposed previously is not accurate. In the revision, we removed it and changed to "role of dust source temperature changes". Detailed results and responses are shown in the following. We wish the current version of manuscript could give more convincing results and arguments.

Major comments:

The Indian summer monsoon is primarily driven by the thermal contrast between land and ocean (Wang et al., 2000). The up troposphere meridional temperature gradient south of Tibetan Plateau is one of the key controls of the Indian summer monsoon (Li and Yanai, 1996). An up troposphere warming over TP tends to increase the meridional temperature gradient, and intensifies the Indian summer monsoon (Wu et al., 2005; Liu et al., 2001). Previous studies show that both DRE and SDE of absorbing aerosols could induce a warming around TP and intensify the Indian summer monsoon (EHP effect), which is in general consistent with the observed relationship. In this study, however, the authors found the Indian summer monsoon is intensified (weakened) associated with cooling (warming) over TP, which is just opposite to previous studies. The authors should carefully check the model settings and give some explanations.

RE: Please see the next two comments for detailed explanations.

The authors found that the dust SDE induced TP warming tends to weakens Indian summer monsoon, which is opposite to what found in Qian et al. 2011. The authors stated that the opposite response is due to the difference in TP warming center distribution. Lau et al. 2010 found that aerosol SDE could produce an "elevated-heat-pump (EHP)" effect and increase the precipitation over Indian in May. Their results is in generally consistent with Qian et al. 2011, although the warming center is over western TP. The authors should provide convincing explanations why the response to dust SDE in this study is contradictory to previous studies.

RE: Yes. From previous studies (e.g., Lau et al., 2010; Qian et al., 2011), it can be obtained that the SDE of black carbon or all absorbing aerosols intensifies the Indian monsoon, which is different from what we simulated for dust. We pointed out that our

study is not contradictory to previous studies because our study merely focused on dust and the spatial distribution of dust is different. In our opinion, the key point is that we can not directly compare our results and previous studies. Thus, we conducted three additional experiments, with a special focus on black carbon, to support our study. The aim is to examine whether the SDE and DRE of black carbon is similar with mineral dust or not.

[Figure]

Figure R2: Spatial distribution of changes in precipitation rates (a, mm day$^{-1}$), surface temperature (b) and 850hPa wind vectors (c, m/s) in May and June induced by snow-darkening effect of black carbon.

The SDE of black carbon warms the surface over western TP only, with a cooling over northern India (Figure R2a). The southerly winds and precipitation over India are significantly larger (Figure R2b, R2c), which indicates that the summer monsoon is intensified during the onset. This is consistent with what were found in Qian et al (2011), which proves that our experiments are not contradictory to previous studies on black carbon.

However, the same model with the SDE of dust (not specifically focused) gives a quite different response of monsoon, which indicates different mechanisms behind

SDE of dust and black carbon. As we proposed, the spatial distributions of dust and black carbon are not similar. The main difference is that black carbon from the industrial countries is generally transport eastwards and scarcely into upwind Central Asia. Central Asia is also covered by snow although far less than TP. As a result, the forcing of black carbon is restricted to western TP but the forcing of dust is over Central Asia and western TP. These differences in surface warming by dust and black carbon are also simulated in the experiments by NASA Goddard Earth Observing System Model (Yasunari et al., 2014).

The SDE of dust induces significant warming over western TP and Caspian Sea in Central Asia (Figure 5c), which leads to two cyclonic anomalies over these two regions (Figure 6a). These two cyclonic anomalies intensify the northern branch of Indian monsoon westerly, allowing more dry air from Central Asia penetrating into the monsoon region. But the southern branch of the monsoon westerly is decreased with the associated anticyclonic anomaly over Arabian Sea and India, which weakens the moisture transport from oceans in the south.

There is no doubt that the important role of TP temperature change in Indian monsoon development (Li and Yanai, 1996; Wu et al., 2005; Liu et al., 2001), as the reviewer said. We did not argue against it, however, the role of dust source temperature (not mentioned before) is highlighted from our results. We put Figure R2 in the supplement and largely revised the manuscript (Page 6 Lines 14-17, 21-26; Page 7 Lines 3-8, 13-16, 24-28; Page 8 Lines 4-6, 9-11, 29-35; Page 9 Lines 6-9, 18-29; Page 10 Lines 23-28). Some paragraphs in the original text are deleted. We do not show these intensive revisions here and please see the text.

The dust DRE impacts on Indian summer monsoon is also inconsistent with previous studies, and the results are difficult to understand. The authors found dust DRE could induce a significant cooling over TP, and the cooling is attributed to snow-albedo feedback. The dust AOD is very small over TP (less than 0.05), which implies very weak DRE. How such small DRE produce strong snow-albedo feedback over TP? The feedback processes should be detailed explained. More confusing thing is that the Indian summer monsoon (ISM) is intensified associated with the TP cooling, which is similar to the response induced by TP warming (Lau et al., 2006). The authors simply explained it as a response to downward motion right over TP, which is not convincing. Please provide detailed explanations and supportive reference.

RE: Our black carbon experiments show that the DRE of black carbon can strengthen the monsoon, consistent with Lau et al. (2006). A surface warming over western TP is simulated (Figure R3a). The warming is also over northern India although it is not significant. This effect strengthens the southwesterly winds over the Arabian Sea and moisture transport from ocean (Figure S2b) and the precipitation is intensified over the Arabian Sea and southern India (Figure S2c). These results support that the warmer TP intensifies the monsoon, agreeing with traditional viewpoints (Li and Yanai, 1996; Wu et al., 2005; Liu et al., 2001).

However, comparing the responses to dust and black carbon under the same

model and experiment design, we found that the summer monsoon during the onset is both intensified no matter whether it is warming or cooling over the TP. Since the TP cooling is unlikely to intensify the monsoon (at least few evidence support it). We agree it and ascribe the monsoon strengthening to the warming over Arabian Peninsula/Middle East, which also gains strong support from previous researches (Vinoj et al., 2014; Jin et al., 2014; Solomon et al., 2015).

The Arabian Peninsular warming (Figure 5d) induces a local cyclonic anomaly (Figure 6b). The northern branch of monsoon westerly is remarkably reduced in its intensity across the southern slope of the TP, the Persian Gulf and northern Arabian Peninsula (Figure 6b). The southern branch of Indian monsoon westerly over Arabian Sea is simulated to be stronger, which intensifies the water vapor transport from oceans. Although the TP cooling tends to weaken the monsoon, the Middle East warming overacts and induces a stronger monsoon instead. From an observation study, the heating and intensified high pressure cell over Arabian Peninsula is proved to be an important factor affecting the onset of Indian monsoon (Zhang et al., 2014). Thus, based on the new results, we do not emphasize the role of TP and propose the role of dust source temperature in the revision (Page 6 Lines 18-19, 29-33; Page 7 Lines 8-11, 17-18, 28-30; Page 8 Lines 19-21, 29-35; Page 9 Lines 10-14; Page 10 Lines 9-16, 23-28). Figure R3 is also put in the supplement. Some paragraphs in the original text are deleted. We do not show these intensive revisions here and please see the text.

[Figure]

Figure R3: Spatial distribution of changes in precipitation rates (a, mm day-1), surface temperature (b) and 850hPa wind vectors (c, m/s) in May induced by direct radiative effect of black carbon.

For the simulated large cooling over TP, we only found in our analysis that the snow albedo feedback amplifies the response of temperature to small dust forcing. Certainly, we can not deny the possible role of other feedbacks. In addition, compare to that of black carbon, the DRE of dust on temperature is largely uncertain (Kok et al., 2017).

In this study, the CAM4 was run with prescribed climatological SST and sea ice. The SST response to aerosol forcing (slow response) is not taken into account. Many previous works showed that the slow response can play a dominant role in the total response of Indian summer monsoon to aerosol forcing (Ganguly et al., 2012). Many previous studies investigated dust impacts on ISM with coupled simulations (e.g. Qian et al, 2011). It could be a possible reason why the monsoon response is opposite to previous studies. Thus, the authors should run coupled simulations and make a

comparison with current results.

RE: Thanks for the comments. We agree with the reviewer that the slow response of ocean may make the response more complicated. However, due to the limited time of final response phase, it is difficult for us to conduct addtional coupled model simulations, which are always integrated for hundreds of years for quasi-equilibrium. More importantly, three atmospheric GCM experiments focused on black carbon were conducted to support our arguments. The results strongly support the distinct forcing of black carbon and dust on the Indian monsoon via different mechanims, as we said in responses above.

In the revision, we cited the references and emphasized the possible role of slow ocean processes (Page 4 Lines 27-29): "Due to the limit of calculation resource, we only conducted atmospheric model experiments in this study and coupled ocean-atmosphere model experiments are not included. Actually, slow ocean response can play a dominant role in the response of Indian summer monsoon to aerosol forcing (Ganguly et al., 2012)."

This study investigates the dust impacts on Indian summer monsoon. However, only the dust effect during the monsoon onset periods (May and June) is investigated. The Indian summer monsoon is from June to August (or September). Please show the monsoon response in July and August, for the dust concentration is still high in Indian at that time (Gu et al., 2016). The response of ISM could be quite different in July and August, for dust DRE impacts could be more important at that time. Only with an examination of the response in entire monsoon period, the title of this manuscript could be appropriate.

RE: Thanks. The response of monsoon during its mature period (July-September) to DRE and SDE of dust is also important. However, these changes in the precipitation and low-high level circulation are similar but complicated, and also not as significant as those during the onset, possibly because the monsoon onset is more sensitive to radiative and temperature changes. Lots of previous studies indicated the sensitive responses of monsoon onset to external forcing, e.g., the sensible heat changes over TP or to its southwest (e.g., Li and Yanai, 1996; Wu and Zhang, 1998; Wu et al., 2012), in agreement with our study. To be accurate, as the reviewer commented, we revised our title to "Indian summer monsoon onset" in the revision.

Other comments:
Page 1, Line 2:"have" should be "has".

RE: Corrected.

Page 4, Line 6-7:Please give more explanations on "snow-darkening and direct radiative feedbacks". Does it mean the permit of dust snow-darkening and direct

radiative effects in simulations? What is the meaning of feedback?

RE: Yes. We meant the snow-darkening and direct radiative effects are considered in the experiments. We revised the sentence (Page 4 Lines 10-11).

Page 5, Line 13: Please provide references for the two branches of Indian summer monsoon.

RE: A reference is added here (Wu et al., 2012).

Page 5, Line 25: If EXP1-EXP2 equals to the impacts of dust SDE, please use the dust SDE in the rest of manuscript for consistency. So do the cases for EXP2-EXP3.

RE: We used SDE and DRE instead.

Page 5, Line 26: Please clarify the definition of "Indian monsoon area".

RE: We specified the region (10-25°N, 65-100°E) in the text (Page 6 Line 1).

Page 5, Line 28: Indian summer monsoon lasts from June to August. Please show the precipitation change in July and August, as well.

RE: As we responded previously, we changed the title to "... Indian summer monsoon onset". Thus, in the revision, we still showed the monsoon response in May and June.

Page 6, Line 9:Why dust SDE induces significant cooling over Tibetan Plateau? The dust AOD is very small over Tibetan Plateau.

RE: We think here the reviewer means DRE (not SDE). In this paper, the DRE-induced cooling over Tibetan Plateau is explained by the snow-albedo feedback. We do not find the important contributions from other processes in our analyses. More importantly, based on the new results, we do not emphasize the role of TP in the revision. Thus, we turn our eyes on the new-proposed role of dust source temperature and give detailed explanations on this point.

Page 6, Line 15:Why is the southern branch of the monsoon westerly significantly decreased?

RE: We found that surface temperature becomes warmer over most Asia, which responds to the SDE. The most obvious warming is found over western TP where the surface snow cover is larger. Another significant warming center is around Caspian Sea in Central Asia also with certain snow covers at this time.Following the temperature changes, a significant cyclonic anomaly is simulated over western TP and there is also a cyclonic anomaly around the Caspian Sea. These two cyclonic

anomalies tends to intensify the northern branch of Indian monsoon westerly, allowing more dry air from Central Asia penetrating into the monsoon region. However, the southern branch of the monsoon westerly is significantly decreased due to the associted anticyclonic anomaly over Arabian Sea and India.

We emphasized it in the revision (Page 6 Lines 21-26): "In the SDE-induced difference, a significant cyclonic anomaly is simulated over western TP and to its west there is also a cyclonic anomaly around the Caspian Sea (Figure 6a), following the surface temperature changes (Figure 5c). These two cyclonic anomalies tends to intensify the northern branch of Indian monsoon westerly, allowing more dry air from Central Asia penetrating into the monsoon region. However, the southern branch of the monsoon westerly is significantly decreased with the associated anticyclonic anomaly over Arabian Sea and India, which weakens the moisture transport from oceans in the south."

Page 8, Line 10: Please show the dust snow forcing (outputted by SNICAR), dust deposition (dry and wet), and dust concentration in snow over TP and their seasonal variation. A comparison with previous studies (e.g., Qian et al., 2011) is also needed.

RE: We analysed the suggested variables and the results are shown in Figure R4. It is clearly seen that the dust deposition flux and concentration in top snow layer reach its peak in boreal spring. The forcing (as shown by changes in surface radiation, snow cover fraction and albedo) due to SDE of dust is maximal during April-June, the vital period for snow melting. The peak of dust forcing lags the deposition by about one month, which indicates the memory effect of snow processes. The seasonality of dust deposition and dust snow forcing is similar with previous studies, supporting that our experiments are reasonable. Due to the change of our emphasis, these changes over TP are not added to the revision.

[Figure]

Figure R4: Total dust deposition fluxes and mass of dust in top snow layer in EXP1 experiments and the changes in surface radiative fluxes, snow cover fractions and surface albedo due to SDE of dust averaged for the TP region (70-90°E, 30-45°N)

Page 8, Line 23: Please explain the feedback.

RE: We meant the snow-albedo feedback here based on our analyses.

Page 8, Line 23: Dust aerosols could absorb both shortwave and longwave radiative fluxes. Why the longwave radiative flux change is negative?

RE: From our results, only the net longwave forcing for column atmosphere is negative. The reason is that the warmer atmosphere as a black body emits more longwave radiation, which exceeds over the absorbed amount. Previous studies also showed similar features for net longwave radiation change (Albani et al., 2014; Xie et al., 2018).

Page 8, Line 34: The dust AOD is very small over TP (less than 0.05), which implies

very weak DRE. How such weak DRE produce significant snow cover increase and surface cooling over TP? It could not be simply attributed to snow-albedo feedback.

RE: In this paper, the DRE-induced cooling over Tibetan Plateau is explained by the snow-albedo feedback. We do not find important contributions from other processes in our analyses. More importantly, based on the new results, we do not emphasize the role of TP in the revision. Thus, we turn our eyes on the new-proposed role of dust source temperature and give detailed explanations on this point.

Page 9, Line 19-30: In Lau et al. 2010, they found that TP warming tends to increase the Indian precipitation in May, and the warming center is located at western TP. Their result is consistent with Qian et al. 2010, but different with the results of this manuscript. Explanations are needed here.

RE: Please see Figure R2 and the associated response.

Page 10, Line 12: How could downward motion right over TP induce an upward motion over Indian? Is it noticed any previous studies? Please provide more explanations as well as the references.

RE: Thanks. We revised this assertion because we do not have enough evidence. As we discussed in other responses, the TP cooling and downward motion may be not closely associated with the intensified monsoon and upward motion over India. In the revision, we ascribed the intensified monsoon to the warming over Arabian Peninsula, which gains strong support from previous studies (Vinoj et al., 2014; Jin et al., 2014; Jin et al., 2015; Solomon et al., 2015).

Figures:
Figure 2 and Figure 3 could be put in the supplement, for they are too many figures for this manuscript.

RE: We kept Figure 2 and Figure 3 in the manuscript because we removed several figures in the revision. We will put them in the supplement if the reviewer still feels there are too many figures.

Figure 4:Please use the specific date in figure 4 (e.g. May 1st).
Figure 4:Please specify the regions of precipitation change.

RE: We changed the date and also specified the region (10-25°N, 65-100°E) in the caption.

Figure 5:Please display the precipitation and surface temperature with different color tables.

RE: We used different color bars.

Figure 5 and so on: Please show "SDE" and "DRE" in figure title.

RE: We used "SDE" and "DRE" instead of "EXP1-EXP2" and "EXP2-EXP3", respectively.

Figure 5 to Figure 10: There are too many figures for this part. Decide what is important and put the rest in supplement.

RE: We removed original Figure 7, 10, 13 because we wish to avoid the repeating and also do not emphasize the TP temperature any more. Two new figures, showing black carbon's related results, are added in the supplement.

References:

Ganguly, D., P. J. Rasch, H. Wang, and J.-h. Yoon (2012), Fast and slow responses of the South Asian monsoon system to anthropogenic aerosols, Geophysical Research Letters, 39(18), L18804, doi:10.1029/2012GL053043.

Gu, Y., Xue, Y., De Sales, F., Liou, K. N.: A GCM investigation of dust aerosol impact on the regional climate of North Africa and South/East Asia, Clim. Dyn., 46, 2353–2370, 2016.

Lau, K. M., Kim, M. K., and Kim, K. M.:. Asian monsoon anomalies induced by aerosol direct forcing: the role of the Tibetan Plateau, Clim. Dyn., 26, 855–664, 2006.

Lau, K.-M., Kim, M. K., Kim, K.-M., and Lee, W. S.: Enhanced surface warming and accelerated snow melt in the Himalayas and Tibetan Plateau induced by absorbing aerosols, Environ. Res. Lett., 5, 025204 doi:10.1088/1748-9326/5/2/025204, 2010.

Li, C., and M. Yanai (1996), The Onset and Interannual Variability of the Asian Summer Monsoon in Relation to Land–Sea Thermal Contrast, Journal of Climate, 9(2), 358-375, doi:10.1175/1520-0442(1996)009<0358:TOAIVO>2.0.CO;2.

Liu, X., and M. Yanai (2001), Relationship between the Indian monsoon rainfall and the tropospheric temperature over the Eurasian continent, Quarterly Journal of the Royal Meteorological Society, 127(573), 909-937, doi:10.1002/qj.49712757311.

Qian, Y., Flanner, M., Leung, L., and Wang, W.: Sensitivity studies on the impacts of Tibetan Plateau snowpack pollution on the Asian hydrological cycle and monsoon climate, Atmos. Chem. Phys., 11(5), 1929–1948, doi: 10.5194/acp-11-1929-2011, 2011.

Wu, B. (2005), Weakening of Indian summer monsoon in recent decades, Advances in Atmospheric Sciences, 22(1), 21-29, doi:10.1007/BF02930866.

Wang, B., R. Wu, and K.-M. Lau (2001), Interannual Variability of the Asian Summer Monsoon: Contrasts between the Indian and the Western North Pacific–East Asian Monsoons, Journal of Climate, 14(20), 4073-4090, doi:10.1175/1520-0442(2001)

---

## Referee Report (RR1)

**Review of "Snow-darkening versus direct radiative effects of mineral dust aerosol on the Indian summer monsoon onset: role of dust source temperature changes" by Shi et al.**

In previous version of manuscript, the authors emphasized the role of the Tibetan Plateau in dust aerosol effects on Indian summer monsoon (ISM) onset, which is inconsistent with the findings of previous studies. The authors provided exactly new explanations in the revised manuscript, and the role of dust source temperature changes is emphasized. However, the revised manuscript is still not convincing enough to published on Atmospheric Chemistry and Physics. Please see the details in the major comments.

**Major comments:**

A new concept "dust source temperature" was proposed in the revised manuscript, but its meaning is not clear. I guess it means the surface temperature change in dust source regions. Please provide detailed explanations and make it clear in the manuscript.

The dust SDE could induce a warming in both TP and Central Asia, while the warming due to BC SDE is constrained over TP regions. Such difference in warming distributions could produce different ISM responses, as proposed by the authors. However, both the snow fraction (Figure 2a) and surface radiative forcing (Figure 9e) due to dust SDE is small and insignificant over central Asia, which could not explain the warming there. Thus, the response of ISM to dust SDE could not be simply attributed to the warming in dust source regions. The authors should provide solid evidences to show that dust SDE could induce a significant warming over Central Asia.

Moreover, in observation, strong ISM is found associated with the surface warming in Central Asia (Wang et al. 2000, their Figure 7b), which also does not support the conclusions of the manuscript. It is noticed that BC DRE also produces a warming in both Central Asia and TP (Figure S2a), but the ISM (Figure S2c) response is quite different. All evidences suggest that the weakening of ISM induced by dust SDE could not be simply explained by the warming in dust source regions. The authors should provide convincing explanations and supportive references to prove that a warming in Central Asia and western TP can weaken ISM.

For dust DRE, the authors emphasize the role of surface warming in Arabian Peninsula, which is also difficult to understand. The absorbing aerosols (BC and dust) always reduce the solar flux to surface, cool the surface, but warm troposphere (Vinoj et al., 2014). The net surface radiative effect is very small over Arabian Peninsula (Figure 10f), which could not explain the surface warming there. The authors should provide detailed explanations why dust DRE could induce significantly warming Arabian Peninsula at surface.

The authors said what they found in dust DRE effects also gains strong support from previous researches (Vinoj et al., 2014; Jin et al., 2014; Solomonet al., 2015). Although

previous studies show that dust DRE could intensify ISM, the mechanism is totally different. According to previous studies, the dust DRE could intensify ISM through heating the atmosphere, other than inducing a warming at surface. The authors should carefully check the model results, and make a comparison with previous studies.

The authors proposed that, due to the dust DRE, branch of Indian monsoon westerly over Arabian Sea becomes strong and intensifies the water vapor transport from ocean, which is not true. The stronger westerly is constrained over the coastal regions of Arabian Peninsula (Figure 6b), which may not intensify the water vapor transport to Indian. In summary, how dust DRE intensify ISM is still not clear and needed further analysis.

Overall, the main conclusions of this manuscript are still unclear and inconsistent with previous studies. Thus, the manuscript needs further revisions and resubmitted.

**Other comments:**

Line 34: Duplicated periods.

**References:**

Jin, Q., Wei, J., and Yang, Z.: Positive response of Indian summer rainfall to Middle East dust, Geophys. Res. Lett., 41, 4068–4074, 2014

Solmon, F., Nair, V.S., and Mallet, M.: Increasing Arabian dust activity and the Indian summer monsoon, Atmos. Chem. Phys, 15, 80519–8064, 2015

Vinoj, V., Rasch, P. J., Wang, H., Yoon, J., Ma, P., Landu, K., and Singh, B.: Shortterm modulation of Indian summer monsoon rainfall by West Asian dust, Nature Geosci., 7, 308–313, 2014

Wang, B., R. Wu, and K.-M. Lau (2001), Interannual Variability of the Asian Summer Monsoon: Contrasts between the Indian and the Western North Pacific–East Asian Monsoons, Journal of Climate, 14(20), 4073-4090, doi:10.1175/1520-0442(2001)014<4073:Ivotas>2.0.Co;2.

---

## Referee Report (RR2)

**Review of "Snow-darkening versus direct radiative effects of mineral dust aerosol on the Indian summer monsoon onset: role of temperature change over dust sources" by Shi et al.**

The authors provided detailed answers in the response and made corresponding revisions to the manuscript in the second-round revision. But there still some problems with the main conclusions of the manuscript. Thus, the manuscript needs major revisions before it could be accepted. Please see the detailed comments below.

**Major comments:**

The authors emphasized the role of warming around Caspian Sea in dust SDE impacts. Unlike black carbon, dust SDE can induce a warming around Caspian Sea (central Asia) and weakens Indian summer monsoon, as proposed by the authors. The authors also provided detailed physical explanations and the role of snow fraction reduction was emphasized. But there is barely none dust SDE around Caspian Sea (Figure 9E). The snow darkening effect means the snow can absorb more solar flux with the deposition of absorbing aerosols on snow. The solar flux change (Figure 9E) are very small and insignificant around Caspian Sea, which may not explain the warming there. Moreover, the snow fraction change (Figure R2b, Figure 12b) is also very small around Caspian Sea. Please provide more evidences that the warming around Caspian Sea and Central Asia is due to dust SDE.

In my opinion, there are no distinct differences between spatial distribution of dust and BC SDEs. And the difference in ISM response may not be attributed to their different spatial distributions. Please make a detailed comparison of BC and dust SDE and show there is significant difference between them. The SDE could be directly evaluated by the BC/dust snow forcing outputted by SNICAR.

**Other comments**

Figure 12:Please check the time of figure captions and title.

Figure 12:Please show positive values in the color bar. The color bar could be misleading.

---

## Author Response (AR2)

**Review of "Snow-darkening versus direct radiative effects of mineral dust aerosol on the Indian summer monsoon onset: role of dust source temperature changes" by Shi et al.**

In previous version of manuscript, the authors emphasized the role of the Tibetan Plateau in dust aerosol effects on Indian summer monsoon (ISM) onset, which is inconsistent with the findings of previous studies. The authors provided exactly new explanations in the revised manuscript, and the role of dust source temperature changes is emphasized. However, the revised manuscript is still not convincing enough to be published on Atmospheric Chemistry and Physics. Please see the details in the major comments.

Major comments:
A new concept "dust source temperature" was proposed in the revised manuscript, but its meaning is not clear. I guess it means the surface temperature change in dust source regions. Please provide detailed explanations and make it clear in the manuscript.

RE: Yes. **The "dust source temperature" here means the surface/low-level air temperature over dust source regions, i.e., over Arabian Peninsula and Central Asia.** Given that the role of Tibetan Plateau (TP) temperature is intensively emphasized, that of these dust source temperature changes is not mentioned. For the SDE, increase in surface temperature over Central Asia, together with that over TP, induces a twin of cyclonic anomalies in low troposphere. Compared to the change by TP only, the twin cyclones occupy further west, which induce quite different circulation response over Indian monsoon region. For the DRE, we failed to simulate a TP warming. Our strengthened monsoon is originated from the low tropospheric warming over Arabian Peninsula. The detailed reasons are discussed in the following response. To clarify the statement, in the revision we used "temperature change over dust sources" instead and emphasized this term in the results and discussion.

The dust SDE could induce a warming in both TP and Central Asia, while the warming due to BC SDE is constrained over TP regions. Such difference in warming distributions could produce different ISM responses, as proposed by the authors. However, both the snow fraction (Figure 2a) and surface radiative forcing (Figure 9e) due to dust SDE is small and insignificant over central Asia, which could not explain the warming there. Thus, the response of ISM to dust SDE could not be simply attributed to the warming in dust source regions. The authors should provide solid evidences to show that dust SDE could induce a significant warming over Central Asia.

RE: Thanks for the comments. We checked our results and did further analysis to support the arguments (Figures R1 and R2). From the surface radiative forcing (Figure 9f), the total forcing is indeed statistically significant over Central Asia, although it does not cover all the regions. From Figure R1, we can see the sensible heat from the land contributes a lot to the surface air warming. Over the TP, the sensible heat flux is significantly increased, consistent with previous studies. Over Central Asia, the sensible heat is also increased (smaller than TP but also significant) with similar pattern with the surface air temperature. Then we checked the changes in surface ground temperature to see whether it is in agreement with the sensible heat (Figure R2). Yes, we also found remarkable land warming at Central Asia and it is clearly resulted from the decrease in

snow cover. From a perspective of energy balance, the sensible heat release from land is responsible for the surface air warming over Central Asia. **The physical loop is as follows: snow covers decrease (Figure R2b) -> ground temperature increases (Figure R2a) -> sensible heat increases (Figure R1a) -> surface air temperature increases (Figure 5c).** Thus, the surface warming over Central Asia is valid and it is clearly from the decreased snow cover and increased sensible heat flux, at least in our experiments. These differences in surface warming by dust and black carbon are also simulated in the experiments by NASA Goddard Earth Observing System Model (Yasunari et al., 2014). In their figure 2a, we can see that "SDE is found to produce significant surface warming over broad areas in mid latitudes, with dust being the most important contributor to the warming in central Asia and the western Himalayas...", as they stated in the abstract.

The westward expanded warming areas due to dust, compared to those due to black carbon, finally lead to different responses of Indian monsoon onset. **Here, we should emphasize that our results do not deny the importance of TP temperature proposed in previous study for black carbon (Qian et al., 2011).** We totally agree that the response of TP temperature is vital for the Indian summer monsoon. **Our results merely promotes the complexity of monsoon response to temperature pattern. Central Asia is one of the dust sources and also covered by snow. If a perturbance indeed occurs over this region, it may modulate the response of monsoon to TP warming, as we shown in this study.** We added a paragraph in the discussion to emphasize this point.

We put Figure R1 and R2 in the revision and largely revised the manuscript (Page 1 Line 13-16; Page 7 Line 34-35; Page 8 Line 1-7; Page 8 Line 17-19; Page 8 Line 34-35; Page 9 Line 1-10; Page 10 Line 7-11). We do not show these intensive revisions here and please see the text.

[Figure]

Figure R1: Changes in surface sensible and latent heat fluxes in May and June induced by snow-darkening effect (a, b) and direct radiative effect of dust (c, d), respectively. Oblique lines indicate differences significant at 95% confidence level. Yellow line shows the profile of Tibetan Plateau above 2500 m.

[Figure]

Figure R2: Changes in surface ground temperature in May and June (a) and snow cover fraction during March to June (b) induced by snow-darkening effect. Oblique lines indicate differences significant at 95% confidence level. Yellow line shows the profile of Tibetan Plateau above 2500 m.

Moreover, in observation, strong ISM is found associated with the surface warming in Central Asia (Wang et al. 2000, their Figure 7b), which also does not support the conclusions of the manuscript. It is noticed that BC DRE also produces a warming in both Central Asia and TP (Figure S2a), but the ISM (Figure S2c) response is quite different. All evidences suggest that the weakening of ISM induced by dust SDE could not be simply explained by the warming in dust source regions. The authors should provide convincing explanations and supportive references to prove that a warming in Central Asia and western TP can weaken ISM.

RE: Thanks. We do not agree that these two evidence is contraditory to our proposed response of Indian summer monsoon to SDE of dust. From Figure R3 (Figure 7b in Wang et al. 2000), we can see that stronger monsoon corresponds to the warming over western TP and Pakistan/Afghanistan regions (green triangle). However, this warming pattern is different from that induced by dust SDE, in which another warming center is around Caspian Sea (Figure R5). Simply speaking, **the Pakistan/Afghanistan warming center is very near the western TP warming, which exerts similar effects on monsoon with TP warming that driving the monsoon southwesterly winds to directly blow towards Indian subcontinent (Figure 8c in Wang et al 2000). But the warming around Caspian Sea is in further west and drive southwesterly winds to Arabian**

**Pennisula (Figure 6a)**. From Figure R4, as the reviewer mentioned, the black carbon DRE also produces a warming in both TP and central Asia. We note that **this warming is located in exactly the same region (Pakistan/Afghanistan) with Wang et al. 2000. This well explains why we simulated a stronger Indian monsoon to black carbon DRE and it is shown that the southwesterly winds is driven directly to India (Figure S2c, similar with Figure 8c in Wang et al. 2000).** The consistency between model and observation support the ability of model on Indian monsoon simulation to external forcing. In this aspect, these two evidence do not contradict but support our arguments.

[Figure]

Figure R3: JJA surface temperature anomalies between strong and weak Indian summer monsoon years (Wang et al., 2000, Figure 7b).

[Figure]

Figure R4: Changes in surface temperature induced by direct radiative effect of black carbon.

[Figure]

Figure R5: Changes in surface temperature induced by snow darkening effect of dust.

For dust DRE, the authors emphasize the role of surface warming in Arabian Peninsula, which is also difficult to understand. The absorbing aerosols (BC and dust) always reduce the solar flux to surface, cool the surface, but warm troposphere (Vinoj et al., 2014). The net surface radiative effect is very small over Arabian Peninsula (Figure 10f), which could not explain the surface warming there. The authors should provide detailed explanations why dust DRE could induce significantly warming Arabian Peninsula at surface. The authors said what they found in dust DRE effects also gains strong support from previous researches (Vinoj et al., 2014; Jin et al., 2014; Solomon et al., 2015). Although previous studies show that dust DRE could intensify ISM, the mechanism is totally different. According to previous studies, the dust DRE could intensify ISM through heating the atmosphere, other than inducing a warming at surface. The authors should carefully check the model results, and make a comparison with previous studies. The authors proposed that, due to the dust DRE, branch of Indian monsoon westerly over Arabian Sea becomes strong and intensifies the water vapor transport from ocean, which is not true. The stronger westerly is constrained over the coastal regions of Arabian Peninsula (Figure 6b), which may not intensify the water vapor transport to Indian. In summary, how dust DRE intensify ISM is still not clear and needed further analysis.

RE: Thanks for the comment. We agree with the mechanism stated by the reviewer. For the DRE, the absorbing aerosols reduce the solar flux to surface and warm the troposphere. These two features of DRE on shortwave radiation flux is widely accepted and also clearly seen in our results (Figure 10e, 10h). **For the net radiation flux (SW+LW), although it may be positive for the total column atmosphere (Figure 10i), the surface forcing is still uncertain (it is not definitely negative for all regions) and it depends highly on the choice of size distributions, optical property and heights of dust layers and et al (e.g., Kok et al., 2017)**. For example, Albani et al. (2014) simulated a positive surface forcing of dust over most Sahara and Arabian Peninsula (Figure 12i in their paper). **However, these uncertainty do not affect our discussion because these circulation changes are actually in the troposphere.** The reason we used surface air temperature is to compare with the SDE directly. Following the comment, we further examined the low troposphere temperature (Figure R6) and found significant warming over Arabian

Peninsula. The low-level warming induces the anomalous low pressure over Arabian Peninsula (Figure R7), which is consistent with the results in Vinoj et al. (2014). In addition, the response of surface temperature is not simply linked to surface net radiative forcing. **The surface temperature might also be controlled by the TOA net forcing over some regions, which is significantly impacted by atmospheric dynamical processes (e.g., convection, Miller, 2012).** Our simulated positive TOA forcing (Figure 10c) is in agreement with surface warming over Arabian Peninsula.

The reviewer stated that it is not true that the branch of monsoon westerly over Arabian Sea becomes strong and intensifies the water vapor transport from ocean. Although the differences in wind vectors are not statistically significant over some regions, we can still see that a clear southwestward wind anomaly is simulated over Arabian Sea, from southern Indian Ocean to Indian subcontinent (Figure R8 top). Then we calculated the moisture flux and found that **the intensified moisture flux are remarkable (Figure R8 bottom) over nearly the whole Arabian Sea**. Thus, the mechanism in our study is that **the DRE of dust warms low level troposphere over Arabian Peninsula and produces a local low pressure anomaly, which drives the moisture from the ocean to India and finally promotes the rainfall.** This mechanism is indeed similar with previous studies (Vinoj et al., 2014; Jin et al., 2014; Solomon et al., 2015). We admit that our previous statement may be not accurate enough, which might lead to misunderstanding from the readers. Follwing this comment, we put the changes in 850hPa temperature, geopotential height and moisture flux in new Figure S1 in the revision and largely revised the discussion on the DRE of dust in order to make the mechanism clearer (Page 1 Line 11-16; Page 6 Line 18-19; Page 6 Line 30-35; Page 8 Line 3-7; Page 8 Line 31-33; Page 9 Line 17-18; Page 10 Line 22-24; Page 10 Line 25-27; Page 11 Line 8).

[Figure]

Figure R6: Changes in 850hPa and 700hPa temperature induced by direct radiative effect of dust. Oblique lines indicate differences significant at 95% confidence level.

[Figure]

Figure R7: Changes in surface pressure and 850hPa geopotential height induced by direct radiative effect of dust. Oblique lines indicate differences significant at 95% confidence level.

[Figure]

[Figure]

Figure R8: Changes in 850hPa wind vectors and moisture flux induced by direct radiative effect of dust. Black vectors indicate differences significant at 90% and 95% confidence levels for top and bottom subfigures, respectively.

Other comments:
Line 34: Duplicated periods.

RE: Revised.

[revised manuscript text omitted]

---

## Author Response (AR3)

**Review of "Snow-darkening versus direct radiative effects of mineral dust aerosol on the Indian summer monsoon onset: role of temperature change over dust sources" by Shi et al.**

The authors provided detailed answers in the response and made corresponding revisions to the manuscript in the second-round revision. But there still some problems with the main conclusions of the manuscript. Thus, the manuscript needs major revisions before it could be accepted. Please see the detailed comments below.

RE: Thanks for the comment.

Major comments:

The authors emphasized the role of warming around Caspian Sea in dust SDE impacts. Unlike black carbon, dust SDE can induce a warming around Caspian Sea (central Asia) and weakens Indian summer monsoon, as proposed by the authors. The authors also provided detailed physical explanations and the role of snow fraction reduction was emphasized. But there is barely none dust SDE around Caspian Sea (Figure 9E). The snow darkening effect means the snow can absorb more solar flux with the deposition of absorbing aerosols on snow. The solar flux change (Figure 9E) are very small and insignificant around Caspian Sea, which may not explain the warming there. Moreover, the snow fraction change (Figure R2b, Figure 12b) is also very small around Caspian Sea. Please provide more evidences that the warming around Caspian Sea and Central Asia is due to dust SDE.

In my opinion, there are no distinct differences between spatial distribution of dust and BC SDEs. And the difference in ISM response may not be attributed to their different spatial distributions. Please make a detailed comparison of BC and dust SDE and show there is significant difference between them. The SDE could be directly evaluated by the BC/dust snow forcing outputted by SNICAR.

RE: Thanks for the comment. The reviewer is right that the surface shortwave radiation change should be visible if the SDE is significant. However, in Figure 9E, the surface shortwave radiation change is not clear enough over central Asia. We did a detailed analysis on this problem and found the reason finally. **The SDE forcing is actually not simultaneously with the response of monsoon onset during May and June. The SDE forcing, which is responsible for the Central Asian warming in May and June, occurs in preceding month April.** Due to the thermal inertia of land, the land/ground temperature is significantly affected by snow cover change a month before. The processes are as follows.

First, we checked the simulated April and May-June (MJ) snow covers (Figure R1) over central Asia and found that the central Asian snow cover fraction in MJ is quite small (also in good agreement with MODIS observation). However, **in April, the central Asian snow cover becomes larger, especially to the west of Caspian Sea.** The changes in snow cover due to SDE is then calculated (Figure R2). It can be seen that, the changes in snow cover are not visible in MJ (Figure R2b) because the total snow cover in the control experiment is limited (few snow to melt). But **the**

**changes in snow cover is statistically significant around Caspian Sea in April (Figure R2c), which could affect the land temperature. Since the thermal inertia of land is larger than atmosphere, the reduction in snow cover in April induces remarkable warming over central Asia in MJ**, as seen in the ground temperature (Figure R2a).

**These SDE-associated decrease in snow cover in April also leads to increased shortwave radiation flux at surface around Caspian Sea**, which is larger than 5W/m$^2$ with a maximum of 25 W/m$^2$ (Figure R3). **These evidence support that the central Asian warming is basically from the SDE.** As the reviewer suggested, we also examined the surface dust-in-snow forcing outputted by SNICAR and found that **the surface forcing does not exist in MJ but is significant in April around Caspian Sea (Figure R4a, b), consistent with snow cover changes.** We admit that the initial surface forcing of SDE over central Asia is not so large as that over western Tibetan Plateau (TP). However, it is still a non-ignorable forcing (larger than 1 W/m$^2$ over most regions around Caspian Sea) and strong enough to induce the simulated statistically-significant changes in snow cover and ground temperature. Hence, the physical link behind the SDE and surface air warming over central Asia is that **SDE forcing in April -> increase in shortwave radiation and decrease in snow cover in April -> increased ground temperature in MJ -> increased sensible heat exchange into surface atmosphere in MJ -> surface air warming in MJ**.

Different from central Asia, the changes in surface forcing and snow cover fractions over western TP are consistently significant in April and MJ (Figure R2b, c; R4a, b), which directly affects the sensible heat exchange from land and atmospheric shortwave radiation and contributes to the local warming.

**The surface forcing of black carbon in snow is different from dust that it is not visible over central Asia both in April and MJ (Figure R5), which indicates that black carbon can not induce central Asian warming.** The SDE-induced warming is restricted to western TP and these different spatial patterns of warming finally lead to opposite responses of Indian monsoon, as we argued in the last response. **In a previous study (Yasunari et al., 2015), they pointed out that the contributions of dust and black carbon to the SDE are different and dust dominates the SDE over Central Asia**, which supports our arguments on the different spatial distributions of SDE of dust and black carbon.

At last, we emphasized again that we do not deny the important role of TP warming in the response of Indian summer monsoon. But our results indicate the possibility of the influence of central Asian temperature in the SDE of dust. If there is indeed a perturbance of dust-in-snow over central Asia, it can significantly modulate the response of Indian monsoon to TP warming. In the revision, we added a paragraph to discuss the delayed response of air temperature over Central Asia to the SDE-induced snow cover changes (Page 5 Lines 20-22; Page 8 Lines 22-24; Page 9 Lines 16-28; Page 11 Lines 23-25).

[Figure]

Figure R1: Snow cover fraction (%) over Asia in CAM4 (a, c) and in Moderate Resolution Imaging Spectroradiometer (MODIS)-retrieved observation (b, d) for May to June, and for April, respectively. Yellow line shows the profile of Tibetan Plateau above 2500 m.

[Figure]

Figure R2: Changes in surface ground temperature in May and June (a) and snow cover fraction during May to June (%, b) and April (%, c) induced by snow-darkening effect. Oblique lines indicate differences significant at 95% confidence level. Yellow line shows the profile of Tibetan Plateau above 2500 m.

[Figure]

Figure R3: Changes in the surface shortwave radiation fluxes during April due to SDE of dust. Oblique lines indicate differences significant at 95% confidence level. Yellow line shows the profile of Tibetan Plateau above 2500 m.

[Figure]

Figure R4: Surface forcing of dust-in-snow during May to June (W m$^{-2}$, a) and April (W m$^{-2}$, b). Yellow line shows the profile of Tibetan Plateau above 2500 m.

[Figure]

Figure R5: Surface forcing of black-carbon-in-snow during May to June (W m$^{-2}$, a) and April (W m$^{-2}$, b). Yellow line shows the profile of Tibetan Plateau above 2500 m.

Other comments

Figure 12:Please check the time of figure captions and title.

Re: We changed the figures. Please see the response below.

Figure 12:Please show positive values in the color bar. The color bar could be misleading.

Re: We showed the positive values in the color bar to avoid the misleading.

[revised manuscript text omitted]